# The dimerization equilibrium of a ClC Cl⁻/H⁺ antiporter in lipid bilayers

Rahul Chadda[1], Venkatramanan Krishnamani[1], Kacey Mersch[1], Jason Wong[1,2], Marley Brimberry[1], Ankita Chadda[1], Ludmila Kolmakova-Partensky[3], Larry J Friedman[3], Jeff Gelles[3], Janice L Robertson[1*]

[1]Department of Molecular Physiology and Biophysics, University of Iowa, Iowa City, United States; [2]Department of Natural Sciences, University of Bath, Bath, United Kingdom; [3]Department of Biochemistry, Brandeis University, Waltham, United States

**Abstract** Interactions between membrane protein interfaces in lipid bilayers play an important role in membrane protein folding but quantification of the strength of these interactions has been challenging. Studying dimerization of ClC-type transporters offers a new approach to the problem, as individual subunits adopt a stable and functionally verifiable fold that constrains the system to two states – monomer or dimer. Here, we use single-molecule photobleaching analysis to measure the probability of ClC-ec1 subunit capture into liposomes during extrusion of large, multilamellar membranes. The capture statistics describe a monomer to dimer transition that is dependent on the subunit/lipid mole fraction density and follows an equilibrium dimerization isotherm. This allows for the measurement of the free energy of ClC-ec1 dimerization in lipid bilayers, revealing that it is one of the strongest membrane protein complexes measured so far, and introduces it as new type of dimerization model to investigate the physical forces that drive membrane protein association in membranes.

**\*For correspondence:** janice-robertson@uiowa.edu

**Competing interests:** The authors declare that no competing interests exist.

## Introduction

Membrane protein folding involves the favorable association of non-polar protein interfaces amidst an excess of similarly non-polar lipid solvent (*Popot and Engelman, 1990*). Surprisingly, the thermo-dynamic forces driving this assembly remain poorly understood, due to a shortage of experimental systems where reversible equilibrium association can be observed in membranes. Dimerization models of single-pass transmembrane (TM) helices have provided a tractable system for free-energy measurements in detergent micelles (*Fleming et al., 1997*; *MacKenzie and Fleming, 2008*), and recently, in lipid bilayers (*North et al., 2006*; *Chen et al., 2010*; *Hong et al., 2010*; *Yano et al., 2011*). However, the relatively small change in solvent accessible surface area upon dimerization (*MacKenzie et al., 1997*) limits their potential to study protein-specific van der Waals interactions and lipid-solvent-dependent effects, the two driving forces hypothesized to be major players within the membrane environment (*Popot and Engelman, 1990*; *White and Wimley, 1999*; *Bowie, 2005*). Alternatively, studying the dimerization of multi-TM helix membrane proteins offers a new approach, as these interfaces are much larger, and each subunit is expected to adopt a stable, functional fold that could constrain the reaction to a two-state equilibrium. One example that appears particularly well suited is the homodimeric ClC-ec1 Cl⁻/H⁺ antiporter native to *Escherichia coli* (*Maduke et al., 1999*; *Dutzler et al., 2002*). This is a 50-kDa membrane protein that dimerizes via a membrane embedded, non-polar interface lined mainly by isoleucines and leucines (*Figure 1—figure supplement 1A*). Our previous work showed that insertion of bulky tryptophans at the interface destabilized the dimer in detergent, while preserving functional 2:1 Cl⁻/H⁺ transport and structural fold as

**eLife digest** Cells are encapsulated by membranes that form a barrier between the inside of the cell and the outside world. These membranes primarily consist of fatty molecules called lipids, but they are also packed with proteins such as ion channels and transporters that control which molecules pass in and out of the cell. It is proposed that membrane proteins fold spontaneously inside of the cell membrane to adopt the structures that allow them to carry out their function. While it is generally understood why proteins fold spontaneously when they are in water, it is less clear why this occurs for membrane proteins in the oily cell membrane.

Proteins fold into specific shapes because of favorable interactions between different surfaces of the molecule and because the folded structure increases the number of states available to the protein and the surrounding environment. Measuring a quantity known as the "free energy" reports the net balance between these thermodynamic factors and is the first step towards understanding why a membrane protein adopts its particular stable structure in the cell membrane. However, few experimental systems are suitable for studying this reaction in membranes.

One important group of membrane proteins that offers a new approach to studying this question is the ClC family of channels and transporters. These are large proteins that in *Escherichia coli* bacteria and other organisms have only been observed as a "dimer" made up of two identical ClC molecules (or "monomers"). It is however possible that within cell membranes, the ClC transporter proteins switch between their dimer and monomer forms. Reducing the number of proteins in the membrane could reveal these monomers, and allow the free energy associated with forming a dimer from two monomers to be measured.

Chadda et al. diluted *E. coli* ClC protein from natural cell membranes into large synthetic membranes to reduce the number of proteins far below the amount normally seen in cells. Examining the membranes using a technique called single molecule fluorescence microscopy revealed that ClC does exist as monomers when present in low amounts in a membrane. Furthermore, measuring the free energy associated with forming a dimer showed that ClC is one of the strongest membrane protein dimers measured so far.

Chadda et al. also found that ClC is more likely to be in its monomer form if a bulky amino acid called tryptophan is added to the interface at which two ClC molecules bind to each other. Future studies will investigate the mechanism that underlies this change in stability. Ultimately, ClC could serve as a model system to study the forces associated with protein assembly in membranes and answer fundamental questions about membrane protein folding.

ascertained by X-ray crystallography (*Robertson et al., 2010*). Furthermore, a distant ClC homologue, ClC-F, shows equilibrium exchange in detergent micelles (*Last and Miller, 2015*), raising the possibility of free-energy measurements of ClC dimerization in membranes.

Here, we measure the equilibrium dimerization free energy of ClC-ec1 in lipid bilayers by diluting the protein into large membranes and measuring the change in the monomer vs. dimer population. If the system is in a state of dynamic equilibrium, then diluting the protein in the lipid bilayer will shift the population to the monomeric state. To measure the proportion of monomers and dimers as a function of density, we incubated Cy5-labeled ClC-ec1 in large 10 μm diameter multilamellar vesicles (MLVs), then measured the probability that 1, 2, or more Cy5-labeled subunits are captured into extruded liposomes by single-molecule photobleaching analysis using total internal reflection fluorescence (TIRF) microscopy. This approach measures the monomer-dimer equilibrium in the MLV state at the point of extrusion, and as such reports the statistical mechanical dimerization free energy in the lipid bilayer.

The sensitivity of the single molecule approach allows for inspection of the protein at sub-biological densities, i.e. less than one subunit per typical cell membrane. With this technical development in hand, we determined that equilibrium ClC-ec1 subunit exchange occurs on a laboratory timescale and that the reaction follows an equilibrium dimerization isotherm as a function of protein density in the membrane. This allows for the measurement of the free energy of ClC-ec1 dimerization in lipid bilayers and the change in free energy due to tryptophan substitutions at the dimerization interface.

This work introduces ClC-ec1 as an ideal platform for investigating thermodynamic driving forces underlying membrane protein assembly in membranes.

## Results

### Single-molecule co-localization microscopy of ClC-ec1 in 2:1 POPE/POPG liposomes

In a previous study, we showed that tryptophan substitutions I201W and I422W at the dimerization interface of ClC-ec1 (*Figure 1—figure supplement 1*) yield a functionally folded, monomeric form of the transporter in lipid bilayers (*Robertson et al., 2010*). To set up the system for fluorescence studies, we moved a partially buried cysteine to a more accessible position, C85A/H234C that allows for quantitative labeling by Cy5-maleimide without impacting stability (*Figure 1—figure supplement 2*). For simplicity, we will refer to this single-exposed cysteine construct as WT, and tryptophan substitutions as W (I422W) and WW (I201W/I422W).

To measure the dimerization reaction of ClC-ec1 in lipid bilayers (*Figure 1A*), we reconstituted Cy5-labeled protein in 2:1 POPE/POPG lipids at different mole fractions ($\chi$ subunit/lipid) and freeze/thawed the proteoliposomes to produce ~10 µm diameter multilamellar vesicles (MLVs) (*Pozo Navas et al., 2005*). This creates a model of an infinite bilayer where subunits can exchange with one another, even at low dilutions (*Figure 1B*). After equilibration, the membranes are fractionated by extrusion (*Figure 1C*), forming small liposomes that are then imaged on a TIRF microscope (*Figure 1D*) for single-molecule photobleaching analysis (*Figure 1—figure supplement 3D*). With single-molecule sensitivity, we can count the number of subunits captured into each liposome (*Figure 1E*) and determine the probability distribution of fluorescent protein occupancy in liposomes (*Figure 1F*). This approach extends another single-vesicle fluorescence method that examines the behavior of membrane proteins in individual liposomes (*Mathiasen et al., 2014*). However, in our case, we are not studying the state of the protein in the final proteoliposome. For example, a spot that bleaches in two steps could represent a dimer or two independent monomers trapped in the same vesicle. Our approach ignores this ambiguity as it measures the probability that two fluorescent subunits were captured in the same vesicle, reporting the proximity of subunits at the point of extrusion of the large membranes. Therefore, the liposome extrusion step captures the monomer-dimer equilibrium in the prior MLV membrane state and ignores any changes in protein density or lipid composition that might arise during the extrusion process.

Using two-color TIRF microscopy on ClC-ec1-Cy5 reconstituted into fluorescent liposomes (*Figure 1—figure supplement 3*), we find that nearly all Cy5 spots co-localize with liposomes, demonstrating the fidelity of reconstitution. We then counted photobleaching steps in all the imaged Cy5 spots. While the high signal and low background allows for counting of up to nine discrete steps, we found that single-, double- and $\geqq$ three-step photobleaching probabilities, $P_1$, $P_2$ and $P_{3+}$ are sufficient to describe the monomer vs. dimer population in the large MLVs:

$$P_1 = \frac{\# \ of \ spots \ photobleaching \ in \ 1 \ step}{total \ \# \ of \ spots} \tag{1}$$

$$P_2 = \frac{\# \ of \ spots \ photobleaching \ in \ 2 \ steps}{total \ \# \ of \ spots} \tag{2}$$

$$P_{3+} = \frac{\# \ of \ spots \ photobleaching \ in \ \geq 3 \ steps}{total \ \# \ of \ spots} \tag{3}$$

Under ideal experimental conditions, i.e. dilute conditions and 100% fluorescent labeling yield, a single or a double photobleaching step corresponds to a monomer or dimer respectively. In reality, labeling is imperfect, so and $P_2$ include additional states such as singly labeled dimers or two monomers, respectively. In addition, to quantify the dimerization reaction, one must examine the liposome occupancy as a function of increasing protein density in the membrane, a condition that increases the chance of randomly trapping independent subunits in the same liposome, whether associated as dimers or not. The probability for fluorescent subunit capture follows an apparent Poisson

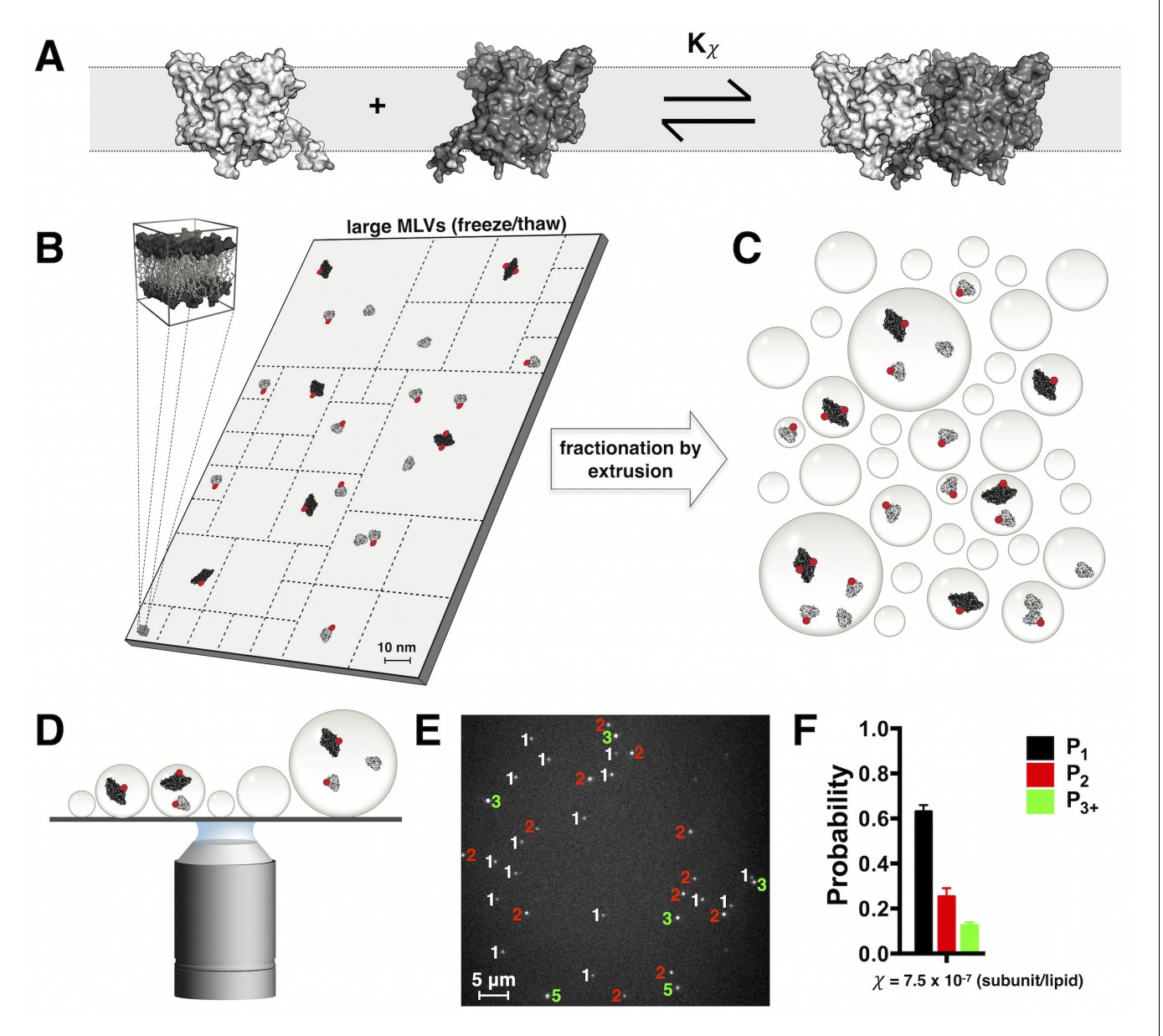

**Figure 1.** Quantifying ClC-ec1-Cy5 monomers vs. dimers in lipid bilayers by subunit capture into liposomes and single-molecule photo-bleaching analysis. (**A**) Cartoon depicting the equilibrium dimerization reaction of ClC-ec1 in lipid bilayers. $K_\chi$ is the mole fraction ($\chi$ subunit/lipid) equilibrium constant. (**B**) Scaled cartoon of a 75 nm × 150 nm area of lipid bilayer, depicting the population of ClC-ec1 distributed as monomers (grey) and dimers (black). ClC-ec1 monomers are ~5 nm across. To allow for subunit exchange at low densities, samples are equilibrated in a large membranes obtained by repeated freeze/thaw cycles to form large multilamellar vesicles (MLVs). Red circles represent Cy5 fluorophores conjugated to subunits with $P_{Cy5}$ ~ 70% labeling yield. (**C**) To quantify the monomer vs. dimer populations in the MLV state, membranes are fractionated (dashed lines in (**B**)) by extrusion which captures subunits into liposomes. The statistics of subunit capture into liposomes follows a Poisson distribution that depends on the overall density, liposome size distribution and population stoichiometry. (**D**) Subunit occupancy in liposomes is determined by examining protein-occupied liposomes on a single-molecule TIRF microscope and carrying out photobleaching analysis. (**E**) Image of Cy5-labeled ClC-ec1 in 2:1 POPE/POPG liposomes. Numbers indicate the observed photobleaching steps for each fluorescent spot (1-white, 2-red, $\geqq$ 3-green). (**F**) Photobleaching probability distribution for a ClC-ec1 sample reconstituted at $\chi = 7.5 \times 10^{-7}$ subunits/lipid (0.1 µg/mg ClC-ec1/lipid). $P_1$, $P_2$ and $P_{3+}$ indicate probabilities of observing single, double and $\geqq$ 3 step photobleaching steps, respectively.

The following source data and figure supplements are available for figure 1:

*Figure 1 continued on next page*

*Figure 1 continued*

**Source data 1.** Excel file including data and statistical analysis presented in *Figure 1* and *Figure 1—figure supplements 1–3* including Ellman's cysteine reactivity data, Cy5 labeling yields, $Cl^-$ transport rates, functional $F_{0,Cl}$, subunit/lipid mole fraction quantification, Fraction of protein co-localized with liposomes and $F_0$, the fraction of unoccupied liposomes measured from co-localization microscopy.

**Figure supplement 1.** Design of ClC-ec1 constructs for the study of reversible dimerization in membranes by fluorescence methods.

**Figure supplement 2.** Stability and function of Cy5 labeled ClC-ec1.

**Figure supplement 3.** Co-localization of ClC-ec1-Cy5 and AF488 labeled 2:1 POPE/POPG liposomes measured by single-molecule TIRF microscopy.

distribution that depends on: (i) the size distribution of the liposomes, (ii) the mole fraction subunit density in the lipid bilayer, (iii) Cy5 labeling yields and (iv) the monomer-dimer equilibrium in the membrane at the time of extrusion. The first three factors must be known and corrected for in order to properly extract information about the monomer-dimer reaction across a wide range of mole fraction densities.

To calculate this correction, we used the cryo-EM liposome size distribution reported by Walden et al. (*Figure 2—figure supplement 1E*) (*Walden et al., 2007*) and spectrophotometrically determined Cy5 labeling yields (*Figure 2—figure supplement 1A*) to simulate the capture of non-interacting monomer into the extruded liposome population, i.e. the ideal monomer photobleaching probability distribution $P^M_n$ (*Figure 2A*), where n refers to the number of observed photobleaching steps. Measurement of the subunit/lipid mole fraction after freeze/thaw and extrusion show that the experimental mole fraction is 50% of the original reconstituted (*Figure 1—figure supplement 2E,F*). From here on, $\chi$ refers to the observed mole fraction, and it is this value that is considered in all of the simulations. At dilute conditions, i.e. densities less than $\chi = 2 \times 10^{-6}$ subunit/lipid (0.2 µg/mg reconstitution density, see *Table 1*), $P^M_1$ is constant and close to one, reflecting single subunit occupancy in the liposomes. The simulation also predicts a non-zero $P^M_2$ that arises from the small amount of non-specific labeling at sites other than the cysteine, resulting in the occasional double-labeled subunit. For $\chi > 2 \times 10^{-6}$ to $3.8 \times 10^{-4}$, $P^M_1$ decays to 0 as $P^M_{3+}$ increases to 1, reflecting the increase in random co-encapsulation of monomers into the same liposome. This has been described before as 'artifactual togetherness' (*Tanford and Reynolds, 1976*; *Fleming et al., 1997*; *Stanley and Fleming, 2005*) and must be corrected for to extract the true dimerization reaction. Next, we simulated a population of non-interacting dimers to obtain the ideal dimer photobleaching probability distribution $P^D_n$ (*Figure 2B*). $P^D_1$ and $P^D_2$ are comparable across the entire protein density range, arising from the fact that our experimental labeling yield is ~70%, leading to an equal probability of dimers labeled with one Cy5 vs. two Cy5 in the single-molecule range (*Figure 2—figure supplement 1D*). For $\chi > 7.5 \times 10^{-6}$, $P^D_1$ and $P^D_2$ decrease as the protein density increases reflecting the growing probability of liposomes with more than two dimers, described by $P^D_{3+}$. To assess the dynamic range for this approach, we performed a chi-squared analysis of the monomer vs. dimer probability distributions at each mole fraction value. The distributions were statistically different ($p \cong 0.0001$) for all mole fraction values except for the highest measured density, $\chi = 3.8 \times 10^{-4}$ or 50 µg/mg ($p > 0.05$). Therefore, the photobleaching probability distributions can distinguish between the monomer vs. dimer state across 5 orders of magnitude: $\chi = 7.5 \times 10^{-10}$ to $7.5 \times 10^{-5}$.

## ClC-ec1 follows a monomer to dimer reaction as a function of protein density in the membrane

The single-molecule photobleaching method allows us to explore extremely dilute densities within the membrane with no loss of signal. At lower densities, the number of fluorescent spots in a field of view decreases, which can easily be compensated for by increasing the number of imaged fields to maintain similar counting statistics. We used this approach to investigate whether experimentally measured $P_1$, and $P_{3+}$ reflect the equilibrium population of ClC-ec1 monomers and dimers in the membrane. If the system is in dynamic equilibrium, then the fraction of dimers out of all subunits will depend on the mole fraction density, $\chi$ (subunit/lipid), but not the path followed to reach this density. We tested this by comparing the photobleaching data obtained using two distinct methods of

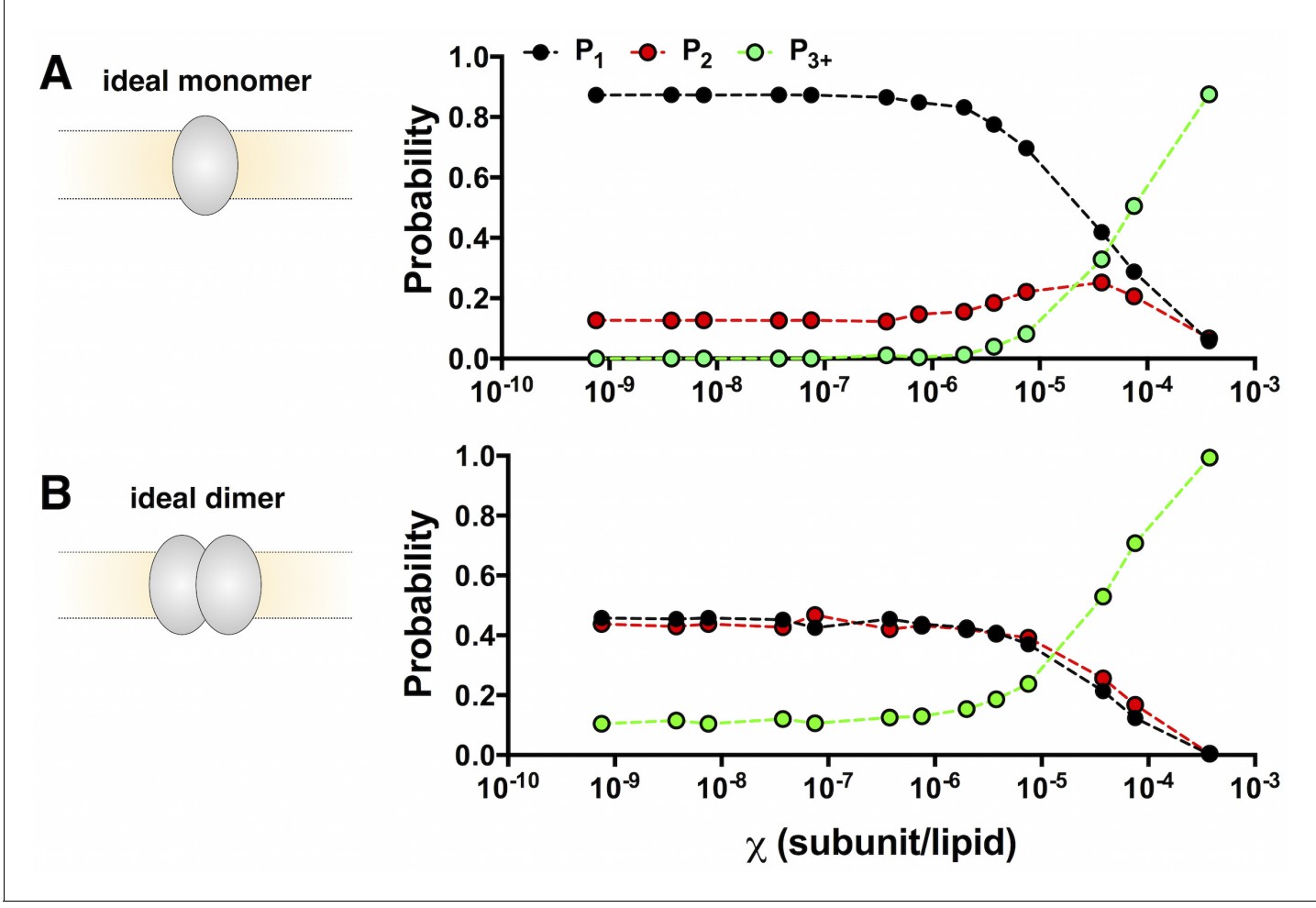

**Figure 2.** Calculation of the ideal monomer and dimer photobleaching probabilities in 0.4 µm extruded 2:1 POPE/POPG liposomes. (A) $P^M_{1,2,3+}$ calculated for $\chi = 7.5 \times 10^{-10}$ to $3.8 \times 10^{-4}$ subunits/lipid for an ideal, non-interacting monomer. The simulation uses the liposome radius probability distribution from Walden et al. (*Walden et al., 2007*), and experimental fluorescent labeling yields $P_{Cy5} = 0.72$ and $P_{non-specific} = 0.14$ (*Figure 2—figure supplement 1*). The appearance of noise in the simulated curves arises from the stochastic nature of the simulation. (B) The ideal, non-interacting dimer photobleaching probabilities $P^D_{1,2,3+}$ simulated with the additional constraint that dimers (~10 nm) are excluded from liposomes r < 25 nm. The excluded radius was estimated from fitting the calculated fraction of unoccupied liposomes, $F_0$, to the experimental data from co-localization imaging (*Figure 1—figure supplement 3*). Chi-squared analysis shows that the monomer and dimer distributions are statistically significant for all $\chi$ values (p < 0.0001) except for $3.8 \times 10^{-4}$ subunits/lipid (p = 0.24).

The following source data and figure supplement are available for figure 2:

**Source data 1.** Excel file including data and statistical analysis presented in *Figure 2* and *Figure 2—figure supplement 1* including the ideal monomer and ideal dimer photobleaching distributions (walden liposomes, $P_{Cy5} = 0.72$, $P_{ns} = 0.14$, bias = 4) distribution, chi-squared analysis, pooled protein labeling data.
**Figure supplement 1.** Probability distributions used to calculate $P^M_n$ and $P^D_n$.

setting the final mole fraction, one that starts with the protein in monomeric form, and another that starts with the protein as a dimer.

In the first method, monomeric W-Cy5 was reconstituted at $\chi = 7.5 \times 10^{-8}$, $7.5 \times 10^{-7}$ and $7.5 \times 10^{-6}$ subunit/lipid by dialysis (*Figure 3A*). Size exclusion chromatography of W in n-Decyl-β-D-Malto-pyranoside (DM) micelles shows a mixture of monomers and dimers upon purification (*Robertson et al., 2010*); however, the protein rapidly dissociates to the monomeric form as revealed by an immediate re-run of the eluted protein (*Figure 1—figure supplement 1D*).

**Table 1.** Lookup table for converting between membrane density units.

| $\rho$ (µg/mg) | $\chi_{Reconstitution}$ (subunits/ lipid) | $\chi_{Observed}$ (subunits/ lipid) | $\chi^*$ (subunits/lipid) | $(\chi^*)^{-1}$ (lipids/ subunit) | $\rho^*_{area}$ (subunits/ nm$^2$ bilayer) | $(\rho^*_{area})^{-1}$ (nm$^2$ bilayer/ subunit) | Box (nm × nm) |
|---|---|---|---|---|---|---|---|
| 0.0001 | $1.5 \times 10^{-9}$ | $7.5 \times 10^{-10}$ | $3.8 \times 10^{-10}$ | 2,657,101,103 | $1.3 \times 10^{-9}$ | 797,130,331 | 28,233 |
| 0.0005 | $7.5 \times 10^{-9}$ | $3.8 \times 10^{-9}$ | $1.9 \times 10^{-9}$ | 531,420,221 | $6.3 \times 10^{-9}$ | 159,426,066 | 12,626 |
| 0.001 | $1.5 \times 10^{-8}$ | $7.5 \times 10^{-9}$ | $3.8 \times 10^{-9}$ | 265,710,110 | $1.3 \times 10^{-8}$ | 79,713,033 | 8928 |
| 0.005 | $7.5 \times 10^{-8}$ | $3.8 \times 10^{-8}$ | $1.9 \times 10^{-8}$ | 53,142,022 | $6.3 \times 10^{-8}$ | 15,942,607 | 3993 |
| 0.01 | $1.5 \times 10^{-7}$ | $7.5 \times 10^{-8}$ | $3.8 \times 10^{-8}$ | 26,571,011 | $1.3 \times 10^{-7}$ | 7,971,303 | 2823 |
| 0.05 | $7.5 \times 10^{-7}$ | $3.8 \times 10^{-7}$ | $1.9 \times 10^{-7}$ | 5,314,202 | $6.3 \times 10^{-7}$ | 1,594,261 | 1263 |
| 0.1 | $1.5 \times 10^{-6}$ | $7.5 \times 10^{-7}$ | $3.8 \times 10^{-7}$ | 2,657,101 | $1.3 \times 10^{-6}$ | 797,130 | 893 |
| 0.2 | $3.0 \times 10^{-6}$ | $2.0 \times 10^{-6}$ | $8.0 \times 10^{-7}$ | 1,328,551 | $2.5 \times 10^{-6}$ | 398,565 | 631 |
| 0.5 | $7.5 \times 10^{-6}$ | $3.8 \times 10^{-6}$ | $1.9 \times 10^{-6}$ | 531,420 | $6.3 \times 10^{-6}$ | 159,426 | 399 |
| 1 | $1.5 \times 10^{-5}$ | $7.5 \times 10^{-6}$ | $3.8 \times 10^{-6}$ | 265,710 | $1.3 \times 10^{-5}$ | 79,713 | 282 |
| 5 | $7.5 \times 10^{-5}$ | $3.8 \times 10^{-5}$ | $1.9 \times 10^{-5}$ | 53,142 | $6.3 \times 10^{-5}$ | 15,943 | 126 |
| 10 | $1.5 \times 10^{-4}$ | $7.5 \times 10^{-5}$ | $3.8 \times 10^{-5}$ | 26,571 | $1.3 \times 10^{-4}$ | 7971 | 89 |
| 50 | $7.5 \times 10^{-4}$ | $3.8 \times 10^{-4}$ | $1.9 \times 10^{-4}$ | 5,314 | $6.3 \times 10^{-4}$ | 1594 | 40 |

$\rho$ is the reconstituted mass density of µg of ClC-ec1 subunits per mg of 2:1 POPE/POPG lipids.

$\chi_{Reconstitution}$ is the reconstituted mole fraction of ClC-ec1 subunits per lipid.

$\chi_{Observed} = \chi_{Reconstitution} * 0.50$, determined from protein to lipid quantification assays.

$\chi^*$ is the reactive mole fraction calculated as $\chi_{Observed}/2$, assuming that the reaction occurs between oriented subunits in the membrane.

$\rho^*_{area}$ is the reactive mole density, subunits per bilayer area, using $SA_{lipid} = 0.6$ nm$^2$.

Box – square root of $(\rho^*_{area})^{-1}$

Bolded values indicate the observed dynamic range of the photobleaching approach.

Photobleaching analysis on extruded liposomes derived from these membranes (*Figure 3C*) shows that at the lowest density, the probability distribution resembles the ideal monomer distribution (*Figure 2A*) and as $\chi$ increases, the distributions approach the ideal dimer distribution (*Figure 2B*) with $P_1$ decreasing as $P_2$ and $P_{3+}$ increase.

In the second method, we started with W-Cy5 already reconstituted in lipid bilayers at high density, $\chi = 7.5 \times 10^{-6}$ subunit/lipid (*Figure 3B*), a condition where W exists almost entirely as dimers. This state was confirmed by macroscopic FRET measurements in MLVs showing that W-Cy3 + W-Cy5 and WT-Cy3/Cy5 yield the same dimer FRET signal (*Figure 3—figure supplement 1*). The high-density W-Cy5 membranes were then diluted 100X and 10X by freeze/thaw-mediated fusion with empty vesicles. At 1 day after freeze/thaw, $P_{1-3+}$ (*Figure 3D*) shows no significant difference with the probability distributions obtained with W-Cy5 samples directly reconstituted at the corresponding $\chi$, $7.5 \times 10^{-8}$ and $7.5 \times 10^{-7}$ subunit/lipid, respectively (*Figure 3C*). Measurement of the 100X diluted photobleaching probabilities as a function of time after freeze/thaw shows no change up to 27 days at room temperature (*Figure 3E*). The agreement between the two methods demonstrates that the reaction is reversible and that equilibrium subunit exchange of W-Cy5 occurs during the freeze/thaw fusion step.

Since the equilibration kinetics may be different for different constructs, we conducted the same experiment for WT-Cy5. So far, WT has only been observed to exist as dimers in lipid bilayers, but all experiments conducted have examined the protein at relatively high mole fraction densities $\chi \geqq 7.5 \times 10^{-7}$ subunit/lipid (i.e. $\rho \geqq 0.1$ µg/mg). We reconstituted WT-Cy5 at $\chi = 7.5 \times 10^{-9}$ to $7.5 \times 10^{-6}$ subunit/lipid, up to 100-fold lower than the lowest density studied so far. At the lowest density, there is a significant $P_1$ probability (*Figure 3F*) indicating that WT exists as monomers in the lipid bilayer at low dilutions. Similar to the W-Cy5 data, as $\chi$ is increased, $P_1$ decreases while $P_2$ and $P_{3+}$ increase indicating a conversion from monomers to dimers. Next, high-density samples of WT-Cy5 reconstituted at $\chi = 7.5 \times 10^{-6}$ were diluted 1000X, 100X and 10X with empty vesicles by freeze/thaw-mediated fusion (*Figure 3G*). The diluted probability distributions show no significant

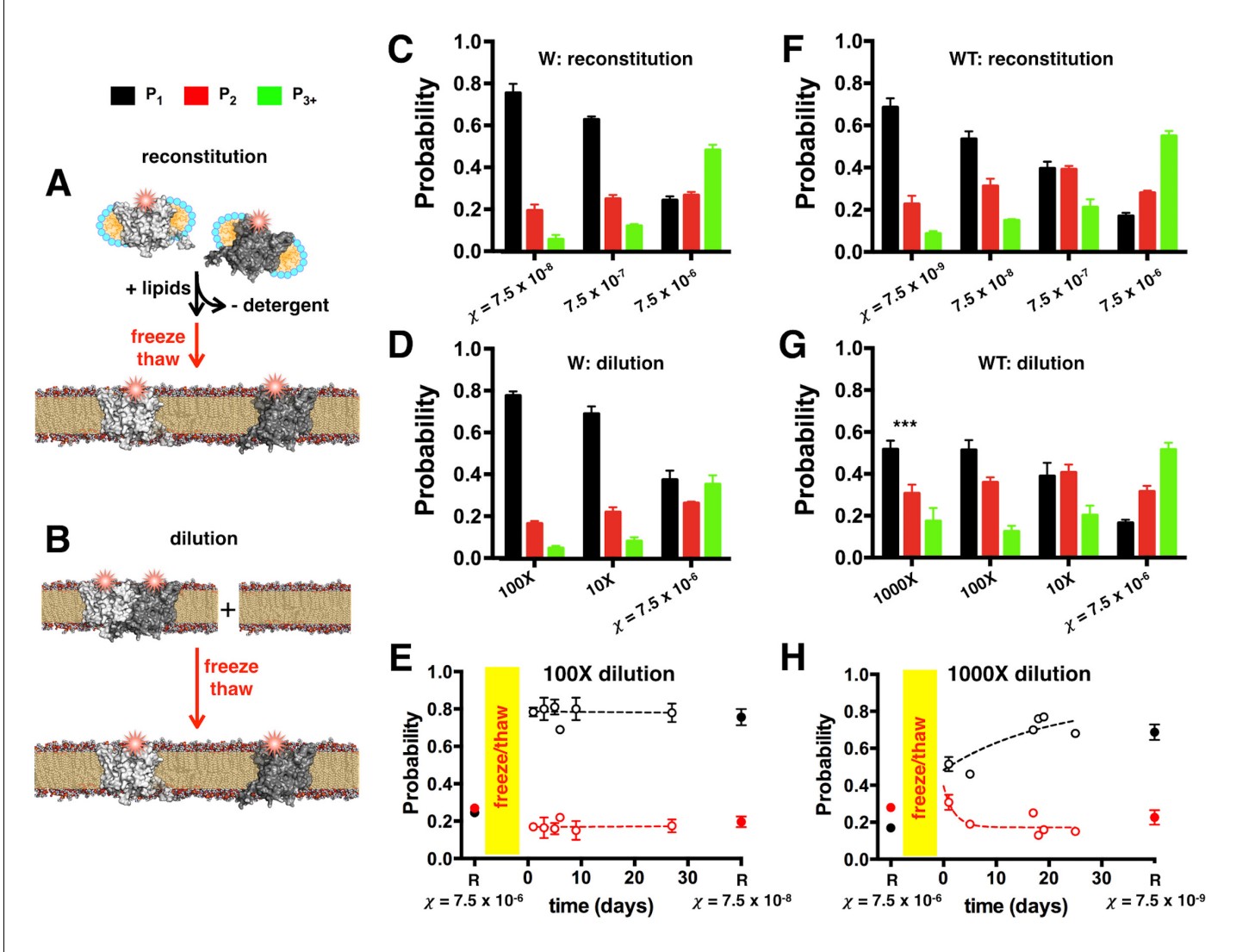

**Figure 3.** ClC-ec1-Cy5 photobleaching probabilities depend on $\chi$ and is path independent. The mole fraction density can be set by two different methods: (A) Reconstitution of ClC-ec1-Cy5 by mixing detergent solubilized subunits with lipids, followed by dialysis to remove detergent resulting in lipid bilayer formation (black arrow). In this case, bilayers are fused together by freeze/thaw (red arrow) and incubated at room temperature prior to extrusion and imaging. (B) Dilution of high-density proteoliposomes by freeze/thaw fusion with empty vesicles, followed by incubation at room temperature prior to extrusion and imaging. (C) Photobleaching probabilities for W-Cy5 reconstituted at $\chi = 7.5 \times 10^{-8}$, $7.5 \times 10^{-7}$ and $7.5 \times 10^{-6}$ subunit/lipid. All distributions are statistically significant by chi-squared analysis ($p < 0.0001$). Data are represented as mean ± SE, n = 3 and 2 counters. (D) Photobleaching probabilities for samples diluted 100X or 10X from samples reconstituted at $\chi = 7.5 \times 10^{-6}$ subunit/lipid and imaged one day after freeze/thaw fusion. Data are represented as mean ± SE, n = 3 and 2–3 counters. Chi-squared analysis shows no significant difference between $\chi = 7.5 \times 10^{-7}$ reconstituted vs. 10X diluted samples, or $\chi = 7.5 \times 10^{-8}$ reconstituted vs. 100X diluted samples ($p > 0.05$). (E) Post-freeze/thaw time course of $P^W_{1-3+}$ probabilities for the 100X diluted sample as a function of incubation time at room temperature. 'R' at $\chi = 7.5 \times 10^{-6}$ subunit/lipid represents the original high-density reconstituted sample prior to dilution (mean ± SE, n = 3 and 2 counters, collected at t = 15, 28 and 72 days). The freeze/thaw process is indicated by the yellow bar. Time course data represent fraction ± SE (n = 3 samples and 2–3 counters, points without error bars represent calculated fraction from a single counter). 'R' at $\chi = 7.5 \times 10^{-8}$ subunit/lipid shows the probabilities for samples reconstituted directly at the corresponding dilution (mean ± SE, n = 2 samples and 2 counters, collected at t = 23 and 71 days). (F) Photobleaching probabilities for WT-Cy5 reconstituted at $\chi = 7.5 \times 10^{-9}$, $7.5 \times 10^{-8}$, $7.5 \times 10^{-7}$ and $7.5 \times 10^{-6}$ subunit/lipid or (D) diluted 1000X, 100X or 10X from samples reconstituted at $\chi = 7.5 \times 10^{-6}$ subunit/lipid, imaged 1 day after freeze/thaw fusion. Data are represented as mean ± SE, n = 3–4 samples and 2 counters. Chi-squared analysis shows no significant difference between $\chi = 7.5 \times 10^{-7}$ reconstituted vs. 10X diluted samples, or $\chi = 7.5 \times 10^{-8}$ reconstituted vs. 100X diluted samples ($p > 0.05$); however, the 1000X diluted sample is significantly different from the $\chi = 7.5 \times 10^{-9}$ reconstituted sample ($p < 0.001$). (E) Post-freeze/thaw time course of $P^{WT}_{1-3+}$ probabilities for 1000X dilution as a function of incubation time at room temperature. 'R' at $\chi = 7.5 \times 10^{-6}$ subunit/lipid designates the high-density sample prior to dilution (mean ± SE, n = 3 and 2 counters, collected at t = 10, 15 and 95 days). Time course data represent

*Figure 3 continued on next page*

*Figure 3 continued*

mean ± SE (n = 3 samples and 2–3 counters). 'R' at $\chi$ = 7.5 × 10$^{-9}$ subunit/lipid represent samples reconstituted directly at the corresponding density, mean ± SE, n = 2 samples and 2 counters, collected at t = 1 and 89 days post-freeze/thaw.

The following source data and figure supplement are available for figure 3:

**Source data 1.** Excel file including data and statistical analysis presented in *Figure 3* and *Figure 3—figure supplement 1* including WT and W dilution data, chi-squared analysis between reconstituted and diluted samples, and FRET data.

**Figure supplement 1.** FRET measurements of WT-Cy3/Cy5 and W-Cy3 + W-Cy5 in lipid bilayers at $\chi$ = 7.5 × 10$^{-6}$ subunit/lipid.

difference compared to the reconstituted distributions except for the 1000X diluted sample corresponding to $\chi$ = 7.5 × 10$^{-9}$ subunit/lipid. Since these data were collected one day after the freeze/thaw process, we continued to image the 1000X diluted WT-Cy5 samples for up to 25 days after freeze/thaw, incubating the samples at room temperature. The photobleaching probabilities of the diluted sample slowly converged to the reconstituted $\chi$ = 7.5 × 10$^{-9}$ subunit/lipid probabilities with a mid-point of 13 days (*Figure 3H*). This demonstrates that the reconstituted WT samples reflect the protein population in a dynamic equilibrium, albeit with higher kinetic stability compared to W, a finding that has been observed with other membrane proteins such as Diacylglycerol kinase (*Jefferson et al., 2013*).

Next, we measured $P_{1\text{-}3+}$ for the three ClC-ec1 constructs reconstituted across a wide range of mole fraction densities: $\chi$ = 7.5 × 10$^{-10}$ to 3.8 × 10$^{-4}$ subunit/lipid. The experimental photobleaching distribution for WW-Cy5, $P^{WW}_{1\text{-}3+}$ (*Figure 4A*) resembles the ideal monomer distribution $P^M_{1\text{-}3+}$ (*Figure 2A*). For W-Cy5 (*Figure 4B*) and WT-Cy5 (*Figure 4C*), the experimental photobleaching distributions $P^W_{1\text{-}3+}$ and $P^{WT}_{1\text{-}3+}$ exhibit three phases: (i) at low densities they mimic the monomer distribution, (ii) as the density increases, there is a gradual approach to the ideal dimer distribution, and finally (iii) at higher densities, $P_1$ and $P_2$ decrease, while $P_{3+}$ increases indicating the capture of multiple subunits in each liposome. WT and W both demonstrate a monomer to dimer transition; however, the reaction of W is shifted along the $\chi$ subunit/lipid axis, indicating the weaker stability of the W dimer.

Using the data in *Figure 4*, we calculated the fraction of dimer ($F_{Dimer}$) across the range of experimental $\chi$ subunit/lipid densities. To take into account that the protein inserts into membranes in two orientations, but only similarly oriented protein participates in the dimerization reaction, we use the reactive mole fraction scale, $\chi^* = \chi/2$. For each value of $\chi^*$, we carry out a least-squares fit of the residual sum of squares ($R^2$) of the experimental $P_n$ probabilities to the $F_{Dimer}$ weighted linear combination of the ideal monomer $P^M_n$ and ideal dimer $P^D_n$ probability distributions from *Figure 2* (*Figure 5—figure supplement 1*). The $F_{dimer}$ vs. $\chi^*$ data was fit to an equilibrium dimerization isotherm (*Figure 5*) to determine the mole fraction equilibrium constants and the mole fraction standard state free energies, using a standard state density of $\chi° = 1$ subunit/lipid. The data for the single tryptophan mutant, W, shows a complete reaction from monomer to dimer, yielding $K\chi^W = 3.7 ± 1.6 ×$ 10$^6$ (best-fit ± standard error (SE)) and $\Delta G°_W = -9.0 ± 0.3$ kcal/mole in 2:1 POPE/POPG lipid bilayers at room temperature. Note that as the density increases beyond $\chi^* = 1.9 × 10^{-6}$, $R^2$ increases indicating that the experimental probability distributions deviate from $P^D_n$ (*Figure 5—figure supplement 2*), due to the observation of more liposomes bleaching in 3+ steps than expected from theory (*Figure 5—figure supplement 1E*). It is expected that the theoretical calculations at high densities will be sensitive to inaccuracies in the liposome size distribution, especially if larger liposomes or multi-lamellarity have been omitted from the distribution. Another possibility is that there is a small amount of non-specific oligomerization at high mole fractions. To account for these deviations, the fits are weighted by $1/R^2$, allowing us to define an observed dynamic range of = 7.5 × 10$^{-10}$ to 3.8 × 10$^{-6}$ subunit/lipid, or 0.0001 to 0.5 µg/mg ClC-ec1/lipid. While WW only shows the initial part of the reaction, this rise occurs within this dynamic range and is thus likely reflecting the onset of dimerization. Fitting the data to a dimerization isotherm yields $K\chi^{WW} = 2.1 ± 0.8 × 10^5$ lipids/subunit and $\Delta G°_{WW} = -7.3 ± 0.2$ kcal/mole. Fitting of WT data yields leads to $K\chi^{WT} = 2.1 ± 0.5 × 10^8$ and $\Delta G°_{WT} = -11.4 ± 0.1$ kcal/mole, although the baseline must be constrained ($Y_0 = 0.07$) since it is not possible to go to lower dilutions where the all-monomer state is expected to be observed in the

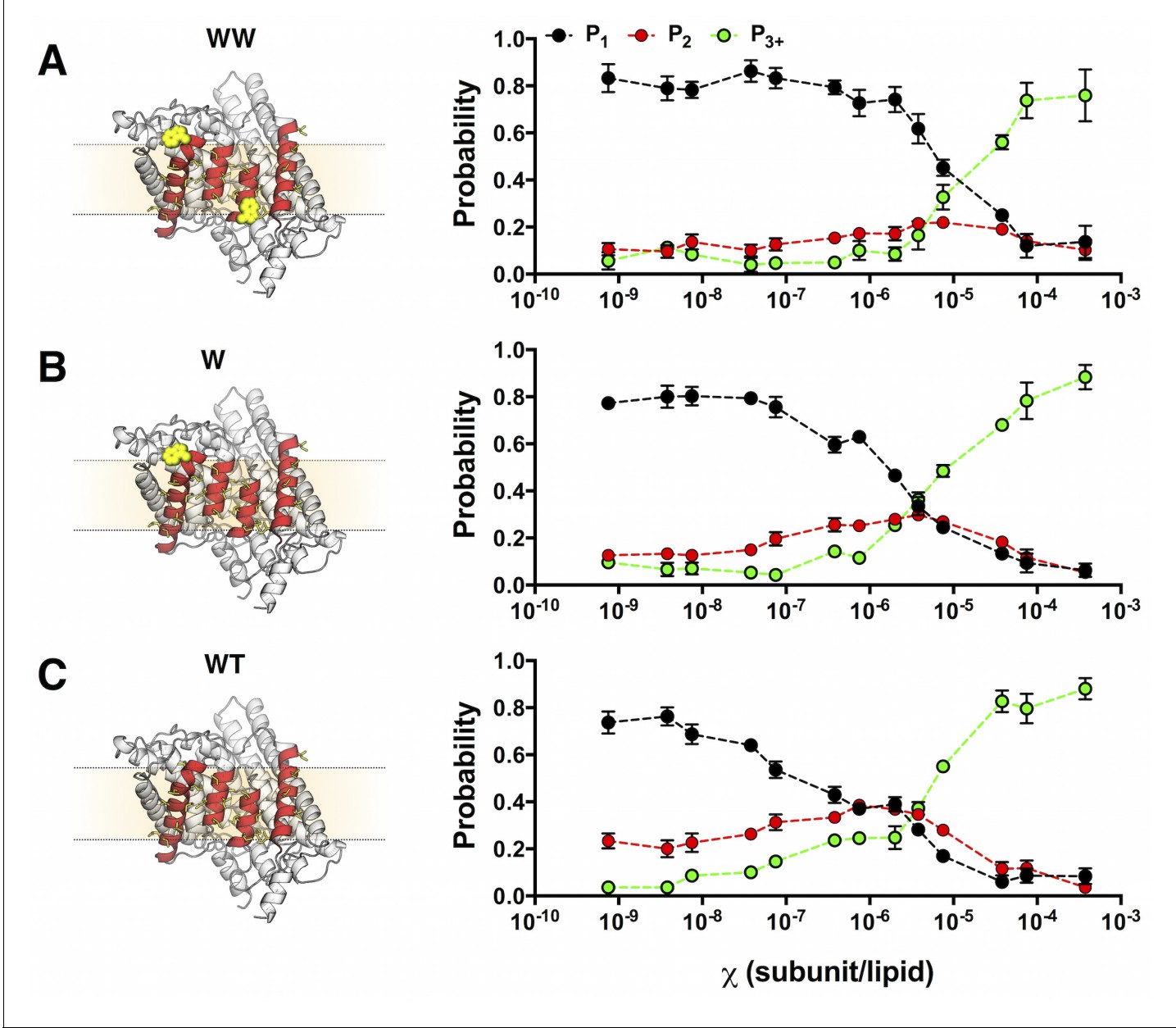

**Figure 4.** Photobleaching probabilities show a monomer to dimer transition that depends on the number of tryptophan residues at the dimerization interface. *Left*, single subunit of (**A**) WW ClC-ec1, (**B**) W and (**C**) WT in the lipid bilayer (beige rectangle, dotted lines) rotated to show the four helices that form the dimerization interface (red) and non-polar residues that line this surface (yellow, licorice). Tryptophan substitutions are shown in yellow VDW representation. *Right*, experimental $P^{expt}_{1-3+}$ at mole fraction densities $\chi = 7.5 \times 10^{-10}$ to $3.8 \times 10^{-4}$ subunit/lipid for (**A**) WT-Cy5, (**B**) W-Cy5 and (**C**) WW-Cy5. Data are reported as mean ± SE (n = 2–3 samples and 2–3 counters).

The following source data and figure supplement are available for figure 4:

**Source data 1.** Excel file including data and statistical analysis presented in *Figure 4* including raw and averaged data for WT, W and WW photobleaching distributions.

**Figure supplement 1.** Robustness of the photobleaching probability distribution.

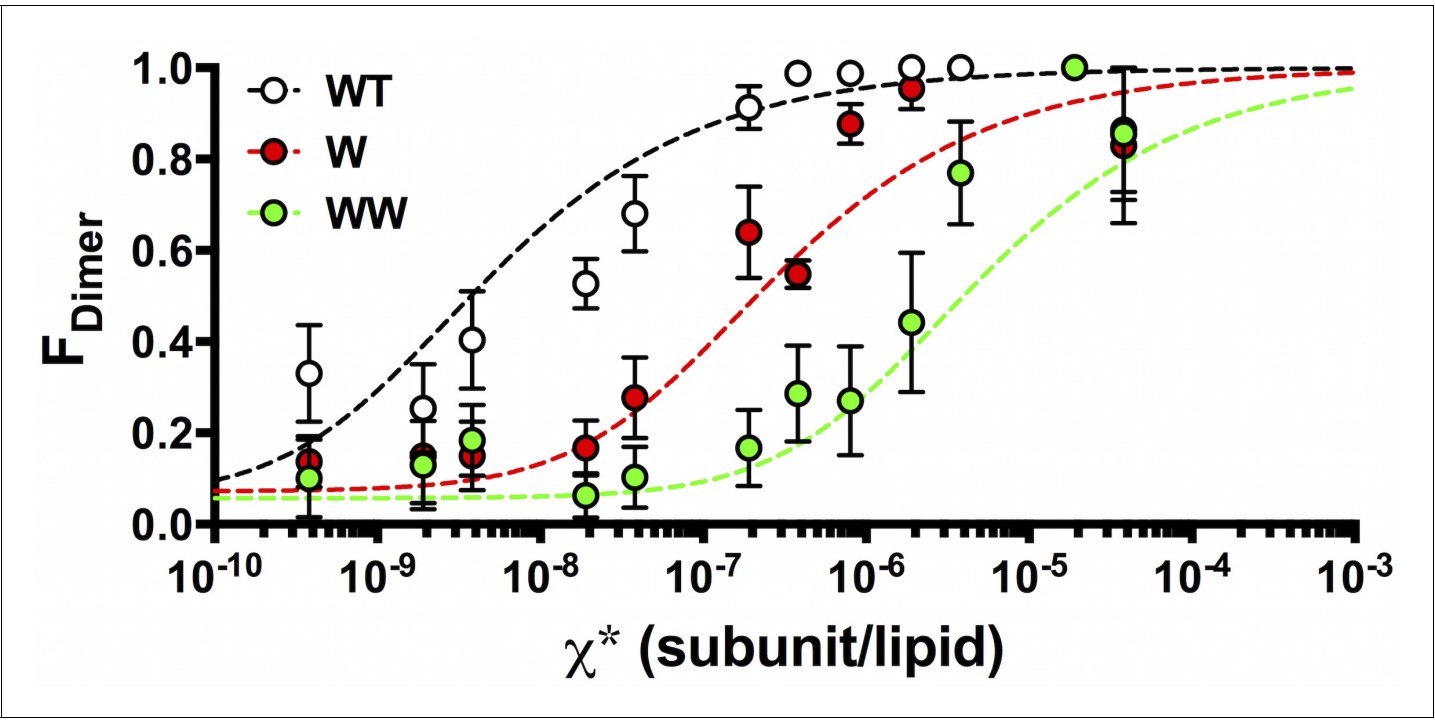

**Figure 5.** $F_{Dimer}$ vs. the reactive mole fraction $\chi^*$. $F_{Dimer}$ is estimated by least-squares fitting of the experimental $P^{expt}_{1-5+}$ photobleaching probabilities to $(1-F_{Dimer})*P^M_{1-5+} + F_{Dimer}*P^D_{1-5+}$, where $P^M_{1-5+}$ and $P^D_{1-5+}$ are the calculated ideal monomer and dimer distributions. The reactive mole fraction, $\chi^*$, is equal to half of the experimental mole fraction ($\chi/2$), assuming that the reaction only occurs between similarly oriented protein in the membrane. Data is shown for WT-Cy5 (black), W-Cy5 (red) and WW-Cy5 (green), with symbols representing mean ± SE (n = 2–3 samples and 2–3 counters). Dotted lines represent best-fits to the equilibrium dimerization isotherm, weighted by the inverse of the minimum residual sum of squares ($R^2$) calculated in the estimation of $F_{Dimer}$ (**Figure 5—figure supplement 1**).

The following source data and figure supplements are available for figure 5:

**Source data 1.** Excel file including data and statistical analysis presented in **Figure 5** including raw and averaged $F_{Dimer}$ data, minimum $R^2$ values, standard error of the estimate (SEE) and $1/R^2$ weights on the fit.

**Figure supplement 1.** Least-squares estimation of FDimer.

**Figure supplement 2.** Sum of squared residuals ($R^2$) as a function of the mole fraction density.

membrane. With this, the results demonstrate that substitution at I422W (W) destabilizes ClC-ec1 dimerization by $\Delta\Delta G_{W-WT}$ = 2.4 ± 0.3 kcal/mole, while the additional tryptophan at I201W (WW) destabilizes the dimer by an additional $\Delta\Delta G_{WW-W}$ = 1.7 ± 0.4 kcal/mole, and $\Delta\Delta G_{WW-WT}$ = 4.1 ± 0.2 kcal/mole overall.

## Discussion

With these results, we can compare and contrast ClC-ec1 with other dimerization models, and take a step toward a generalized understanding of membrane protein stability in the lipid bilayer solvent. In 2:1 POPE/POPG, we find that ClC-ec1 is a high-affinity dimer with $K\chi^{WT\ ClC-ec1}$ ~$10^8$ lipids/subunit, second in stability to GpA in POPC, which has a reported equilibrium constant of $K\chi^{GpA}$ ~$10^9$ (**Hong et al., 2010**). The strength of GpA dimerization is remarkable considering its small 200 Å$^2$ interface, but it has been shown that this stability involves the GxxxG helix-packing motif (**Lemmon et al., 1992**), which allows for backbone flexibility that maximizes VDW packing and hydrogen bonding inside the membrane (**Smith et al., 2002**). ClC-ec1 lacks this specialized motif, so why then is the dimer so stable in lipid bilayers?

To compare between these two very different proteins, we normalize the free energies by the total buried surface area, to obtain a binding efficiency per Å$^2$ of the dimerization interface (*Day et al., 2012*). In this manner, GpA in POPC is highly efficient, contributing -30 cal/mole per Å$^2$, whereas wild-type ClC-ec1 only exhibits -5 cal/mole per Å$^2$ stability. When GpA dimerization was measured in *E. coli* polar lipid membranes, similar to our 2:1 POPE/POPG bilayers, the equilibrium constant shifted to ~5 × 10$^5$ lipids/subunit due to electrostatic destabilization by negatively charged lipids (*Hong and Bowie, 2011*). Even though GpA is slightly weaker than ClC-ec1 in these lipids, its efficiency is still higher, with -20 cal/mole per Å$^2$. Even though ClC-ec1 and GpA reach similar stabilities in the membrane, the dramatic differences in the dimerization efficiencies suggest they do so by different physical mechanisms.

In contrast, dimerization efficiencies of transmembrane helices that do not contain GxxxG motifs have values comparable to ClC-ec1. For example, the Serine Zipper (*North et al., 2006*) is -8 cal/mole per Å$^2$ and poly-LEU-ALA (*Yano et al., 2002*) is -5 cal/mole per Å$^2$, assuming a dimerization interface of 300 Å$^2$. These are often referred to as models of inert helix dimerization, and it is expected that there is minimal conformational change in the helices upon association. In that respect, ClC-ec1 follows an inert surface model with relatively weak interaction efficiency, but it achieves an overall stability comparable to GpA by virtue of its large dimerization interface. The dimer state buries a remarkable 2400 Å$^2$ of protein surface area that would otherwise be interacting with lipids. Decomposition of the free energy into protein-protein, protein-lipid and lipid-lipid terms yields (*Lemmon and Engelman, 1994*; *White and Wimley, 1999*):

$$\Delta G^{\circ}_{dimerization} = \left( \Delta G_{protein-protein} + n\Delta G_{lipid-lipid} \right) - 2n\Delta G_{protein-lipid} \qquad (4)$$

which shows that the free energy is strongly dependent on the number of lipids that solvate the dimerization interface, n. Since n will be relatively large for ClC-ec1 then it is predicted that membrane dependent driving forces, such as hydrophobic mismatch (*Lee, 2004*; *Andersen, 2007*) and changes in lipid entropy (*Lagüe et al., 2001*; *Katira et al., 2016*) could play a larger role in ClC dimerization compared to smaller, single TM-helix dimerization models.

The tryptophan mutagenesis acts as a starting point for quantifying the physical forces associated with protein assembly in membranes. Since the dimerization interface is highly complementary in shape (*Robertson et al., 2010*), addition of a single tryptophan is expected to act as a steric wedge and disrupt many of the VDW interactions between the protein side-chains. However, this ~2 kcal/mole destabilization likely overestimates the VDW contribution since tryptophans will also stabilize the monomeric state by interacting with lipids at this surface. Therefore, this amounts to a relatively small change in dimer stability, indicating that we either maintain many of the VDW contacts in the dimer complex, or that protein-protein VDW interactions are not the major driving force for dimerization. Further investigation of tryptophans as a function of number and position, as well as other residue substitutions, will surely inform on the underlying relationship. In any case, identifying the driving forces that govern ClC-ec1 dimerization will require further experiments that quantify enthalpic and entropic changes while varying protein and lipid dependent variables. Fortunately, the structural integrity and stability of the individual subunits makes investigation of the protein and membrane at different temperatures possible.

The method of single-molecule counting of subunit capture into liposomes addresses many of the challenges previously encountered when studying membrane protein association. Membrane proteins are prone to non-equilibrium aggregation when reconstituted at high densities. Thus, the single-molecule approach enables us to explore low densities where the two-state reaction is expected to dominate. This technique is particularly well suited for studying high-affinity membrane protein complexes that may only show dissociation behavior at sub-biological densities. Our lowest experimental density corresponds to 1 subunit per 50 *E. coli* inner membranes, assuming a 4 μm$^2$ surface area consisting of ~10$^7$ lipids (*Prats and de Pedro, 1989*). Note that the lower limit of the biological mole fraction for *E. Coli* is $\chi$ = 2 × 10$^{-7}$ subunit/lipid, assuming 2 subunits are expressed in the cell. Based on the dimerization isotherm in *Figure 5*, at this mole fraction, 90% of subunits will be found in the dimer form. This means that any reasonable level of expression (10–100 copies per cell) will drive the reaction into a density range where the protein will exist as associated dimers, with negligible probability of observing the dissociated monomeric state. An additional benefit of the photobleaching approach is that it provides checks on the fidelity of reconstitution, as aberrant behavior

such as aggregation would skew the photobleaching distribution toward exceptionally large occupancies. Furthermore, we expect that this method will be useful in the study of other membrane protein systems, even if the structure is unknown, as long as the protein can be purified, quantitatively labeled with fluorophores, and tested for proper function (*Stockbridge et al., 2013*). By combining single molecule approaches with robust membrane protein systems like ClC-ec1, we expect that the measurement of membrane protein association reactions in membranes will no longer be considered a technical challenge, but instead lead to discoveries about fundamental physical driving forces within the lipid bilayer.

## Materials and methods

### ClC-ec1 constructs

All isoforms were inserted into a pASK vector containing a hexa-histidine tag at the C-terminus. Site-directed mutagenesis was carried out by QuickChange (Agilent, Santa Clara, CA) followed by DNA sequencing of the full gene. List of experimental constructs for site-specific labeling: C85A/H234C (WT) MW = 51,997 g/mole, $\varepsilon$ = 46,020 $M^{-1}$ $cm^{-1}$; C85A/H234C/I422W (W) MW = 52,070 g/mole, $\varepsilon$ = 51,700 $M^{-1}$ $cm^{-1}$; C85A/H234C/I201W/I422W (WW) MW = 52,146 g/mole, $\varepsilon$ = 57,410 $M^{-1}$ $cm^{-1}$. List of constructs used to calculate the non-specific labeling: C85A ($WT_{non-specific}$) MW = 52,031 g/mole, $\varepsilon$ = 45,900 $M^{-1}$ $cm^{-1}$; C85A/I422W ($W_{non-specific}$) MW = 52,104 g/mole, $\varepsilon$ = 51,590 $M^{-1}$ $cm^{-1}$; and C85A/I201W/I422W ($WW_{non-specific}$) MW = 52,177 g/mole, $\varepsilon$ = 57,280 $M^{-1}$ $cm^{-1}$. Molecular weight and extinction coefficients calculated using the Peptide Property Calculator at http://biotools/nubic. northwestern.edu/proteincalc.html.

### Protein purification

Expression and purification of ClC-ec1 was carried out as previously described (*Maduke et al., 1999*; *Robertson et al., 2010*). BL21-AI *E. coli* competent cells (Thermo Fisher Scientific, Waltham, MA) were transformed with the plasmid and then 2 L Terrific Broth supplemented with ampicillin was inoculated and grown at 37°C. Protein expression was induced with anhydro-tetracycline at $OD_{600}$ = 1.0. After 3 hr of induction, cells were harvested, then lysed by sonication in buffer supplemented with 5 mM reducing agent TCEP (Tris(2-carboxyethyl)phosphine; Soltec Bioscience, Beverly, MA) and pH adjusted to 7.5. Protein extraction was carried out with 2% n-Decyl-β-D-Maltopyranoside (DM; Anatrace, Maumee OH) for 3 hr at room temperature. Cell debris was pelleted down and the supernatant was run on a 2 mL column volume (CV) TALON cobalt affinity resin (Clontech Laboratories, Mountain View, CA) equilibrated in CoWB/TCEP: 100 mM NaCl, 20 mM Tris, 1 mM TCEP, pH 7.5 with NaOH, 5 mM DM. After binding, the column was washed with 15 CVs of CoWB/TCEP followed by a low imidazole wash of CoWB/TCEP containing 20 mM imidazole (Sigma-Aldrich, St. Louis, MO). ClC-ec1 was eluted with CoWB/TCEP containing 400 mM imidazole, then concentrated in a 30 kDa NMWL centrifugal filters (Amicon, EMD Millipore) to ~500 μL and injected on a Superdex 200 10/30 GL size exclusion column (GE Healthcare, Little Chalfont, UK) equilibrated in size exclusion buffer (SEB): 150 mM NaCl, 20 mM MOPS pH 7.5, 5 mM analytical-grade DM, attached to a medium pressure chromatography system (NGC, Bio-Rad).

### Design of ClC-ec1 constructs for fluorescent labeling

Wild-type ClC-ec1 contains three endogenous cysteines: C85, C302 and C347 (*Figure 1 – supplementary 1B*). While these cysteines can be mutated to yield a 'cys-less' form of ClC-ec1 (C85A/C302A/C347S) that maintain transport function (*Nguitragool and Miller, 2007*), we found that the 'cys-less' substitution on I201W/I422W expresses but results in aggregated protein upon purification. Examining the structure, C85 is partially accessible to the aqueous solution while C302 and C347 are buried within the protein core. We tested whether substituting C85 with alanine alone would be sufficient to minimize background labeling for our fluorescent experiments. We made a construct C85A/H234C, which introduces an aqueous solvent exposed cysteine near the dimerization interface, for specific labeling by Cy5-maleimide (*Figure 1—figure supplement 1B*). A cysteine accessibility assay was used to measure the reactivity of –SH groups present in ClC-ec1 (*Ellman, 1959*; *Riddles et al., 1983*). A 10 mM master stock of Ellman's reagent (DNTB, 5,5'-Dithio-bis (2-nitrobenzoic acid); Sigma-Aldrich) was freshly prepared in reaction buffer (0.1 M sodium

phosphate, 1 mM EDTA, pH 8.0) then diluted to 5 mM with SEB. Reaction of the thiolate anion with DTNB produces 2-nitro-5-thiobenzoate (TNB⁻) that ionizes to $TNB^{2-}$ and absorbs light at 412 nm. $A_{412}$ was monitored by UV-VIS spectroscopy (Nanodrop 2000c, Thermo-Fisher Scientific) for the protein at 10 μM (300 μL) for 5 min to establish a baseline. Following this, 20 μl of the Ellman's reagent working stock was added ([protein] = 9.4 μM, [DNTB] = 313 μM), and the reaction monitored for ten minutes. To estimate the total number of cysteines in the protein, 40 μl of 2% SDS in SEB was added to denature the protein, exposing the buried cysteines C302 and C347 ([protein] = 8.3 μM, [DNTB] = 278 μM, 0.2% SDS), and the reaction was monitored for 45 min, until a steady saturation of $A_{412}$ nm was reached. Absorbance at 412 nm was background subtracted by the absorbance at 750 nm to correct for baseline drift during the measurement. An extinction coefficient of 14,150 $M^{-1}$ $cm^{-1}$ was used to calculate $[TNB^{2-}]$. Addition of Ellman's reagent to C85A/H234C in 5 mM DM shows an instantaneous increase in $A_{412nm}$ signal (*Figure 1—figure supplement 1C*) indicating rapid and specific conjugation with H234C. Addition of 0.2% SDS shows that the internal cysteines are reactive in the SDS denatured state. We calculated the molar ratio of $TNB^{2-}$ produced per ClC-ec1 subunit in DM, and found that there is little reactivity in C85A. All constructs on the C85A/H234C background shows a single reactive thiol in 5 mM DM and a total of three reactive thiols in 5 mM DM + 0.2% SDS, indicating that introduction of tryptophan substitutions do no affect the fold of C85A/H234C in DM micelles (*Figure 1—figure supplement 1D*).

## Fluorescent labeling of protein

Cy5-maleimide dye was obtained as lyophilized powder as either 1 mg (GE Healthcare) or 50 mg (Lumiprobe, Hannover, Germany), stored as 10 mM master stocks (50 μl each) in anhydrous DMSO (Thermo Fisher Scientific) at -80°C. Single-use 5 μl working stocks of 10 mM strength were also prepared and stored at -80°C to avoid multiple freeze/thaw cycles of the fluorophores. Both master and working stocks were stored in boxes in the presence of anhydrous $CaSO_4$ (Drierite, W A Hammond Drierite Co Ltd., Xenia, OH). The fluorophore conjugation reaction was carried out in SEB with 10 μM ClC-ec1 subunits and 50 μM Cy5-maleimide for 12–15 min at room temperature in dark. At the end of the reaction, 100-fold molar excess of cysteine was added to quench the maleimide reaction (from freshly prepared 100 mM stock in SEB, pH adjusted to ~7.5). The 'free' dye was separated from the labeled protein by binding the reaction mixture to a 250 μL cobalt affinity resin column equilibrated with 15 CV CoWB (no TCEP) in a Micro-Bio spin chromatography column (Bio-Rad Laboratories, Hercules CA), washed 15 CV with CoWB and then eluted with 400 mM imidazole in CoWB, manually collecting only the fluorescently labeled protein. To remove the interfering absorbance of imidazole at 280 nm, the labeled protein was added to a 3 mL Sephadex G50 size exclusion column (Sigma-Aldrich) equilibrated in CoWB (no TCEP). The fluorescently labeled protein was eluted after addition 2–2.5 mL of CoWB to the column. The concentration and labeling efficiency of protein calculated from the UV-VIS absorbance spectrum of the sample and $\lambda_{max}$ of ClC-ec1 (280 nm) and Cy5 (655 nm) as follows:

$$[\text{subunit}] = \frac{A_{280} - (A_{\text{fluorophore}} \times CF_{\text{fluorophore}})}{\varepsilon_{\text{subunit}}} \tag{5}$$

$$P_{\text{fluorophore}} = \frac{A_{\text{fluorophore}}}{[\text{subunit}] \times \varepsilon_{\text{fluorophore}}} \tag{6}$$

where $\varepsilon_{\text{subunit}}$ ($M^{-1}$ $cm^{-1}$) is the molar extinction coefficient for the ClC-ec1 isoforms, $A_{\text{fluorophore}}$ is the absorbance of Cy3 or Cy5 at $\lambda_{max}$, $CF_{\text{fluorophore}}$ is the correction factor for fluorophore absorbance at 280 nm ($CF_{Cy3} = 0.08$ $CF_{Cy5}=0.02$) and $\epsilon_{\text{fluorophore}}$ is the molar extinction coefficients for the fluorophore ($\varepsilon_{Cy3} = 1.5 \times 10^5$ $M^{-1}$ $cm^{-1}$ at 565 nm, and $\varepsilon_{Cy5} = 2.5 \times 10^5$ $M^{-1}$ $cm^{-1}$ at 665 nm).

There was no significant difference in labeling yields of the C85A/H234C constructs (*Figure 1—figure supplement 1G*) allowing for the calculation of a pooled average labeling yield of $P_{Cy5} = 0.72 \pm 0.08$ (n = 10). For C85A constructs lacking the reactive H234C, we found that there was no significant difference in labeling and measured a pooled average of $P_{\text{non-specific}} = 0.14 \pm 0.04$ (n = 11), representing the non-specific labeling yield. We tested the various C85A/H234C constructs in 5 mM DM for monomer vs. dimer behavior by size exclusion chromatography (*Figure 1—figure supplement 1C*) and found similar behavior compared to the tryptophan substitutions on the wild-type

background (*Robertson et al., 2010*): C85A/H234C elutes as a dimer, C85A/H234C/I201W/I422W elutes as a monomer, while C85A/H234C/I422W yields a mixture of monomers and dimers upon initial purification, which quickly dissociates into monomers as shown by re-injecting the eluted protein on the size exclusion column. The profiles of un-labelled protein, and protein labeled with Cy5 do not show any significant difference (*Figure 1—figure supplement 2*).

## Fluorescent labeling of lipids

Alexa-Fluor 488 (AF488) SDP ester (Thermo Fisher Scientific) was used to label the primary amine on POPE (1-palmitoyl-2-oleoyl-*sn*-glycero-3-phosphoethanolamine; Avanti Polar Lipids, Alabaster, AL). AF488 was selected as it is spectrally removed from the protein labeling Cy5 channel, and it does not partition into membranes (*Hughes et al., 2014*). Dye stocks were prepared in the same manner described for the cyanine-maleimide stocks. Liposomes were prepared from a 2:1 mixture of POPE and POPG (1-palmitoyl-2-oleoyl-*sn*-glycero-3-phospho-(1'-*rac*-glycerol); Avanti Polar Lipids) as a synthetic mimic of the major phospholipid composition of *E. coli* polar lipid extract. Briefly, a 2:1 mixture of POPE and POPG in chloroform (25 mg/mL) was dried under a continuous stream of $N_2$ gas, then resuspended in 0.5–1 mL pentane and dried again. Labeling buffer (LB): 300 mM KCl, 100 mM $NaHCO_3$ pH 8.3 was added to the lipids for a final concentration of 20 mg/mL (27 mM total lipids: 18 mM POPE, 9 mM POPG). The lipid solution was sonicated in a cylindrical bath sonicator (Avanti Polar Lipids) for 15 min until turbid, then 35 mM CHAPS (Sol-grade; Anatrace) was added and sonication continued until the solution was transparent (30–60 min). AF488 SDP ester was added to the lipid-CHAPS suspension at a final 0.3% mole fraction of total lipids, an amount that allows for visualization of all liposomes as measured by protein co-localization as a function of dye mole fraction. Roughly, the smallest liposomes in the population (r = 10 nm) are estimated to have five dyes assuming a labeling efficiency of 50%. The reaction proceeded at room temperature for 4 hr and then was stopped with 1.5 M Tris (pH 8.0). The lipids were stored at room temperature, in the dark, until fluorescently labeled protein was ready for reconstitution (1–3 hr), and then combined with the Cy5-labeled protein as described in the next section. AF488 fluorescence in the spent dialysis buffer was measured in the fluorometer at $\lambda_{ex}$ = 485 nm, showing no detectable free fluorophore at the end of 48 hr.

## Protein reconstitution

For experiments that did not require fluorescent labeling, lipids were resuspended in either Reconstitution Buffer for functional or microscopy studies (RB-F): 300 mM KCl, 20 mM Citrate pH 4.5 with NaOH, or Reconstitution Buffer for FRET (RB-FRET): 150 mM NaCl, 20 mM Citrate, 10 mM MES, 20 mM Hepes, pH 7.0 with NaOH, with the change to NaCl allowing for the addition of SDS to samples. CHAPS (35 mM) solubilized lipids were combined with protein from 0.0001 to 50 µg ClC-ec1 per 1 mg of lipids, corresponding to $\chi = 7.5 \times 10^{-10}$ to $3.8 \times 10^{-4}$ protein/lipid mole fraction. The protein-lipid-detergent mixture was dialyzed in cassettes (NMWL 10 kDa; ThermoFisher Scientific) at 4°C against 4 L of the appropriate reconstitution buffer for 48 hr with buffer changes every 8–12 hr. After completion of dialysis, the proteoliposomes were harvested from the cassettes, freeze/thawed (see 'TIRF microscopy of proteoliposomes') then stored at room temperature, in the dark until further use.

## Protein/lipid quantification

Quantification of Cy5-labeled protein in proteoliposomes was performed by solubilizing 10 µL of vortexed sample into 190 µL of SEB supplemented with 35 mM CHAPS detergent (Anatrace) and 10% CTAB (Sigma-Aldrich). A standard curve was created alongside each set of samples by mixing serially diluted purified Cy5-labeled ClC-ec1 in the above buffer volume. Protein fluorescence was quantified in a 96-well plate using a Typhoon FLA 9500 Scanner (632nm laser/LPR emission filter permitting light >665nm). In general, the protein labeling yields show little variability between preps, allowing for the determination of amount of protein in each sample well compared to the standard curve.

Quantification of lipids was performed by ashing proteoliposome samples in 8.9 N Sulfuric Acid (Sigma-Aldrich) at 200–215°C for 25 min. Samples were then digested further with concentrated hydrogen peroxide (Sigma-Aldrich) for 30 min at 200–215°C. To each sample milliQ, 2.5%

Ammonium Molybdate (Sigma-Aldrich), and 10% Ascorbic Acid (Macron, Center Valley, PA) was added. Color was developed by heating the samples at 100℃ for 7 min. Samples were placed into a 96-well plate and the absorbance of each sample at 820 nm in a UV-VIS spectrophotometer plate reader. The samples are compared to a standard curve prepared with sodium phosphate dibasic (RPI, Mount Prospect, IL) along side each batch of measurements to determine the molar amount of phosphate (*Fiske and Subbarow, 1925*).

## Functional measurements of chloride transport

To measure $Cl^-$ transport, un-labeled or fluorescent ClC-ec1 isoforms were reconstituted into liposomes in high chloride RB-F at 1 µg/mg, $\chi = 7.5 \times 10^{-6}$. These proteo-liposomes were freeze-thawed seven times, then extruded 21 times through a 0.4 µm polycarbonate membrane (Whatman Nuclepore Track-Etched Membranes). The external buffer was exchanged by passing 100 µL of the liposome sample through a 2.5 mL Sephadex G-50 size exclusion column equilibrated in low-chloride buffer (ExB): 150 mM $K_2SO_4$, 1 mM KCl, 20 mM Citrate at pH 4.5 with NaOH. This sets up a $Cl^-$ gradient, however, efflux of $Cl^-$ ions does not occur because of the opposing potential driving force. A potentiometer to measure $Cl^-$ efflux was setup using silver chloride electrodes (*Walden et al., 2007*). Efflux was measured by the potentiometer in reference to 1 M KCl. For the recording, 1.8 mL of ExB was added to the measurement cell, followed by 15 µL of 10 mM KCl for calibration of the signal. Liposomes (~200 µL) in ExB were added to the measurement cell, and transport was initiated by addition of 1 µM $K^+$ ionophore valinomycin and 2 µM of protonophore FCCP (Sigma-Aldrich). To normalize and compare $Cl^-$ efflux across samples, total $Cl^-$ concentration was measured by breaking the liposomes by adding 40 µl of 1.5 M ß-OG, releasing the remaining chloride trapped inside unoccupied liposomes. The traces were normalized for total $Cl^-$, then fit to a two-component exponential relaxation function to determine $k_{ClC}$ and $F_{Cl,0}$ (*Walden et al., 2007*). Leak from empty 2:1 POPE/POPG liposomes was measured to determine $k_{leak}$.

## Bulk FRET measurements in MLVs

Förster Resonance Energy Transfer (FRET) was measured for upon W-Cy3 + W-Cy5 mixed during reconstitution or co-labeled WT-Cy3/Cy5 in the freeze/thawed MLV state. Fluorescence was measured using a fluorometer with double monochromators on excitation and emission sides to detect fluorescence in highly scattering MLVs (Fluorolog 3–22, Horiba Jobin Yvon, Edison, NJ). Fluorescence emission spectra were collected while exciting the donor ($\lambda_{Ex} = 530$ nm, $\lambda_{Em} = 540$–850 nm, 2 nm slit width, 0.1 s integration time and averaged from 8 to 128 independent sequential scans) or acceptor Cy5 ($\lambda_{Ex} = 640$ nm, $\lambda_{Em} = 650$–850 nm, 2 nm slit width, 0.5 s integration time). FRET was measured by correcting the donor emission spectrum for background fluorescence arising from the buffer and membrane, and for direct excitation of Cy5 when $\lambda_{Ex} = 530$ nm. A correction factor for direction excitation of Cy5 was determined as $0.12 \pm 0.01$ (n = 3) by measuring Cy5-only samples excited at $\lambda_{Ex} = 530$ nm vs. $\lambda_{Ex} = 640$ nm. The reported FRET signal is the area normalization of the FRET-specific Cy5 emission, $I^{Cy5\text{-}FRET}$, over the total emission, $I^{Cy3}$ and $I^{Cy5\text{-}FRET}$:

$$FRET = \frac{\sum_{\lambda=540}^{840} I_{\lambda}^{Cy5-FRET}}{\sum_{\lambda=540}^{840} I_{x}^{Cy3} + \sum_{\lambda=540}^{840} I_{\lambda}^{Cy5-FRET}} \tag{7}$$

The FRET signal was plotted as a function of increasing acceptor to donor ratio ($P_{Cy5}/P_{Cy3}$), fit to a single site binding curve, then normalized by the maximum FRET and fit to the oligomeric model function as described in Fung et al. (*Fung et al., 2009*):

$$Normalized\ FRET = \frac{(R+1)^n - R^n - 1}{(R+1)^n - R^n - 1 + n} \tag{8}$$

where $R = P_{Cy5}/P_{Cy3}$ and n is the number of subunits in the oligomer.

## TIRF microscopy of proteoliposomes

A multi-wavelength single molecule total internal reflection fluorescence microscope was built following the CoSMoS design (*Friedman et al., 2006*; *Larson et al., 2014*). The microscope is equipped with 488 and 637 nm excitation lasers (OBIS, Coherent Inc., Santa Clara, CA) each with

variable attenuators for control of laser power. The lasers were focused onto the back focal aperture of a high numerical aperture 60X objective lens (Olympus TIRF 60X 1.49 NA, Olympus Corporation, Tokyo, Japan) via a 3 mm diameter micro-mirror (MicroMirrorTIRF platform from Mad City Labs Inc., Madison WI). The illuminated area on the coverslip is ~2500 $\mu m^2$. Emitted fluorescence was filtered and focused onto the CCD chip of iXon3 888 EM/CCD camera (Andor Technology Ltd., Belfast, Nothern Ireland).

Glass slides (Gold-seal, 24 × 60 mm no.1.5 thickness) and coverslips (Gold-seal, 25 × 25 mm no. 1.0 thickness) were prepared as follows: slides and cover slips were placed in slide mailers, 5 at a time, then sonicated in presence of 0.1% Micro-90 detergent (Cole-Parmer, Vernon Hills, IL) for 30 min. After sonication, slides and coverslips were rinsed with MilliQ water 10 times to remove all traces of detergent, and then sonicated in presence of absolute ethanol for 30 min followed by rinsing 10 times with MilliQ water. Finally, glass coverslips and slides were sonicated in 0.2 M KOH for 5 min to render the surface hydrophilic. After final rinsing with water the coverslips and slides were stored in MilliQ water in a clean air hood for a period of 7–10 days.

Proteoliposomes harvested after dialysis were freeze-thawed seven times by incubating the samples in a dry ice/ethanol bath for 15 min, followed by incubation at room temperature for 15 min. At this step, the membranes fuse to form large multilamellar vesicles (MLVs) on the order of 10 $\mu$m in diameter (*Pozo Navas et al., 2005*), i.e. $1 \times 10^9$ lipids per lamella. The samples were stored at room temperature with 0.02% $NaN_3$ until imaging, 1–95 days post freeze/thaw (see *Figure 5—figure supplement 2* for specific time points of imaging). The formation of large membranes allows for the investigation of the protein state across a wide span of protein density ( = $7.5 \times 10^{-10}$ to $3.8 \times 10^{-4}$ subunits/lipid), with the lowest value representing roughly two subunits in a liposome lamella of ~300 $\mu m^2$ – i.e., close to infinite dilution. Prior to imaging, the membranes were extruded (Liposo-Fast-Basic; Avestin, Ottawa, Canada) 21 times through a 0.4 $\mu$m Nuclepore membrane to form liposomes for TIRF microscopy with a defined size distribution (*Figure 2—figure supplement 1E*). Between samples, the extruder apparatus was dismantled, flushed profusely with hot tap water, then sonicated in presence of dilute detergent (0.5% Dial dish soap) followed by sonication in deionized $H_2O$ water to maintain extremely low background. After washing, buffer was passed through the extruder and loaded onto the microscope to confirm that there was no contamination between samples. Liposomes were diluted in low-adhesion tubes and allowed to passively bind to glass slide (*Johnson et al., 2002*). The density of fluorescent spots on the glass slide was maintained at 0.02 to 0.09 spots/$\mu m^2$ to minimize overloading (~50–200 spots per field). For experiments at low protein density, the slide was loaded with a high number of liposomes since the probability of protein occupied liposomes was rare. At higher mole fractions, liposomes were serially diluted before loading onto coverslips. Images were acquired at the rate of ~1 frame per second (fps), EM gain set to 300 and laser incident power typically set to 15 $\mu$W for AF488 imaging and 240 $\mu$W for Cy5 imaging in order to obtain long photo-bleaching traces while maintaining good signal to noise for single molecule spots. In each field, Cy5 imaging was carried out before AF488 imaging as we observed the 488 nm laser convert Cy5 to a dark state. After initial liposome-protein co-localization experiments to measure $F_0$ and reconstitution efficiency, samples for protein photobleaching experiments were typically reconstituted with regular POPE/POPG mixture without any fluorescent label on the lipids. All imaging was carried out in RB-F buffer, filtered three times through a 0.22 $\mu$m filter (Millex-GS MCE, Merck Millipore, Billerica, MA).

## Photobleaching analysis

Cy5 blinking (*Ha and Tinnefeld, 2012*) was avoided by imaging the proteoliposomes in the absence of oxygen scavengers. Depletion of oxygen results in longer dwell time in the triplet state which often leads to fluorescent blinking. Triplet-state quenchers, such as Trolox, are often added to the imaging solutions that contain oxygen scavengers to reduce blinking behavior. However, Trolox has been shown to alter lipid bilayer properties by partitioning into membranes (*Alejo et al., 2013*). Instead, we found that avoiding oxygen scavengers altogether allowed us to observe long-lived photobleaching intensity traces of Cy5 without blinking behavior. In addition, the removal of the oxygen scavengers helps to reduce background contamination as we are working with non-passivated clean slides, and we have observed components of the glucose oxidase/catalase oxygen scavenger system demonstrate autofluorescence when bound to the slide. No filtering of images was necessary, as laser power was adjusted for optimal signal to noise while ensuring relatively long photobleaching

traces. Photobleaching data was collected in both unlabeled and AF488-labeled liposomes, showing no significant difference in probabilities (*Figure 4—figure supplement 1*). Image files were analyzed in a MATLAB-based CoSMoS analysis program (*Friedman and Gelles, 2015*). Fluorescent spots were auto-detected based on intensity thresholds selecting 4 × 4 pixel areas of interest (AOIs) around the peak fluorescence. The AOIs were integrated over time; typically 300 frames were acquired at 1 frame per second (fps). Integrated intensity trajectories extracted from these AOIs were manually classified by counting the number of steps before complete photo-destruction of Cy5. To test for subjectivity of counting, 2–3 individuals analyzed the same data set in a blinded approach (*Figure 4—figure supplement 1*). For probabilities calculated from an individual population, we report fraction ± standard deviation (SD), calculated as the binary uncertainty $\sqrt{p(1-p)N}/N$, where p is the probability of a Cy5 spot bleaching in n steps and N is the total number of Cy5 spots in the sample, typically 200–500. In most cases, the probabilities are calculated over 2–6 samples and 2–3 manual counters, with values reported as mean ± standard error (SE) across both samples and counters.

## Calculation of the ideal monomer and ideal dimer photobleaching probability distributions

The capture of ClC-ec1-Cy5 into extruded liposomes is a Poisson process (*Maduke et al., 1999*; *Walden et al., 2007*). Therefore, the probability of observing a certain number of Cy5 fluorophores, $N_{Cy5}$ = 1, 2, 3,... in a liposome is related to the Poisson distribution. However, under experimentally realistic conditions, the actual distribution will differ due to the following reasons: (1) extruded liposome sizes are heterogenous, (2) microscopy visualizes Cy5 and not the protein itself, so un-labeled subunits or subunits with bleached fluorophores are not counted, and (3) there are multiple labeling possibilities for monomers and dimers due to site-specific and non-specific labeling. As long as the liposome size distribution and fluorescent labeling yields are determined experimentally, the expected $P_1$ and $P_2$ photobleaching probabilities for an ideal monomer or dimer can be calculated. We created a MATLAB script (Mathworks, Natick, MA) that simulates the random process of subunit encapsulation given a certain number of subunits, $N_{subunits}$, and liposomes, $N_{liposomes}$, and the cryo-EM determined liposome size distribution, $P_{radius}$, for 0.4 μm extruded liposomes from freeze/thawed *E. coli* polar lipid membranes (composition ~2:1 POPE/POPG) (*Walden et al., 2007*). The experimental mole fraction, $\chi$, is calculated using the following equation:

$$\chi = \frac{N_{subunits}SA_{lipid}}{N_{liposomes}8\pi\sum_r P_{radius}(r)r^2} \tag{9}$$

using a surface area per lipid, $SA_{lipid}$, equal to 0.6 nm$^2$ (*Murzyn et al., 2005*). The simulation follows by creating a matrix of $N_{liposomes}$ for each sub-population with radius r, and randomly inserting each protein species into these liposomes. In our case, we consider only the all-monomer and all-dimer condition, where $N_{protein,M} = N_{subunits}$ and $N_{protein,D} = N_{subunits}/2$. We now discuss how to calculate (i) $P_{label,M}(N_{Cy5})$ and $Plabel,_D(NCy5)$ – the distribution of the fluorescent labeling states for the monomer or dimer, (ii) $N_{liposomes,M}(r)$ and $N_{liposomes,D}(r)$ – the number of liposomes that are accessible to monomers or dimers, and (iii) $N_{protein,M}(r)$ and $N_{protein,D}(r)$ – the total number of monomers or dimers to be inserted into $N_{liposomes,M}(r)$ and .

## $P_{label,M}(N_{Cy5})$ and $P_{label,D}(N_{Cy5})$ - the probability of monomer or dimer labeling with $N_{Cy5}$ fluorophores.

Under realistic experimental conditions, fluorescent labeling is incomplete, and so it is expected that some subunits are invisible to microscopy, i.e un-labeled or bleached. In addition, non-specific labeling leads to multiple fluorescent labels on a single subunit. Using UV-Vis spectrophotometry, we measured labeling yields of $P_{Cy5}$ = 0.72 and = 0.14 (*Figure 2—figure supplement 1A*). Non-specific labeling was measured in ClC-ec1 constructs lacking H234C, reflecting reaction of the maleimide to lysines, the N-terminus (*Hermanson, 2013*) or internal cysteines C302 and C347. Therefore, each subunit can be considered to have two labeling sites, a non-specific position and H234C, where $P_{H234C} = P_{Cy5} - P_{non-specific}$ = 0.58 (*Figure 2—figure supplement 1B*). Let $P(H234C, non-specific)$ define the probability distribution of the possible labeling outcomes of a single subunit with two

labeling sites. Assuming each site can be labeled (*) or un-labeled (○), there are four possible sub-unit labeling states:

$$P(\circ,\circ) = (1 - P_{H234C})(1 - P_{non-specific}) = 0.36 \tag{10}$$

$$P(\circ,*) = (1 - P_{H234C})P_{non-specific} = 0.06 \tag{11}$$

$$P(*,\circ) = P_{H234C}(1 - P_{non-specific}) = 0.50 \tag{12}$$

$$P(*,*) = P_{H234C}P_{non-specific} = 0.08 \tag{13}$$

To calculate the monomer labeling probabilities, we integrate the above distribution as a function of $N_{Cy5} = \{0, 1, 2\}$ and obtain $P_{label,M} = \{0.36, 0.56, 0.08\}$ (*Figure 2—figure supplement 1C*). The dimer probabilities are calculated by multiplying the different combinations of subunit probabilities in equations 10–13, e.g. $P_{label,D}(0) = P(\circ,\circ)^2$, while $P_{label,D}(1) = P(\circ,\circ)P(\circ,*) + P(\circ,\circ)P(*,\circ)$ and so on. The complete list of fluorescent dimer states is shown in *Figure 2—figure supplement 1D*. With this, $P_{label,D} = \{0.13, 0.40, 0.37, 0.09, 0.01\}$ for $N_{Cy5} = \{0, 1, 2, 3, 4\}$. Therefore, for the simulation of protein insertion into liposomes, we consider three different types of labeled monomer species and five types of labeled dimer species distributed as $P_{label,M}$ and $P_{label,D}$.

## $N_{liposomes,M}(r)$ and $N_{liposomes,D}(r)$ – the number of liposomes accessible to monomers vs. dimers

Cryo-EM measurements of 0.4 μm extruded liposomes of membranes made of *E. coli* polar lipids report a heterogeneous population with radii from 10 to 100 nm (*Figure 2—figure supplement 1E*) (*Walden et al., 2007*). We use this experimental measurement of $P_{radius}(r)$ to calculate the total number of liposomes with radius $r$, $N_{liposomes}(r)$:

$$N_{liposomes}(r) = P_{radius}(r) \cdot N_{liposomes} \tag{14}$$

For each bin r, we declare a matrix with $N_{liposomes}(r)$ rows. The number of columns in the matrix depends on the oligomeric species: 3 – for the monomer and 5 – for the dimer, corresponding to the number of $N_{Cy5}$ labeling states.

For the simulation, we must also calculate the number of liposomes that are accessible to monomers, $N_{liposomes,M}(r)$, and those accessible to dimers, $N_{liposomes,D}(r)$. If all liposomes are accessible, then this converges to $N_{liposomes}(r)$. However, if some of the liposomes are inaccessible due to size or curvature limitations (*Mathiasen et al., 2014*), then this number will be different. We model the accessible liposome probability distribution by excluding smaller radius bins from $P_{radius}(r)$ and normalizing the resultant distribution. The fraction of empty vesicles, $F_0$, as a function of ClC-ec1 protein density is measured by co-localization microscopy (*Figure 1—figure supplement 3C*). For WW-Cy5, more liposomes are filled as the density increases, with an exponential decay indicative of a Poisson distribution (*Goldberg and Miller, 1991*; *Walden et al., 2007*; *Stockbridge et al., 2013*). However, WT-Cy5 reaches a plateau of $F_0 = 0.41$ indicating a significant proportion of liposomes that cannot be occupied by the dimer. Considering that the ClC-ec1 dimer is ~10 nm end to end, and that ~40% of liposomes have radii less than 27.5 nm, this suggests that the smaller liposomes are inaccessible to the larger dimer. Therefore, the accessible liposome population is:

$$N_{liposomes,M}(r) = P_{radius,M}(r) \cdot N_{liposomes}(r) \tag{15}$$

$$N_{liposomes,D}(r) = P_{radius,D}(r) \cdot N_{liposomes}(r) \tag{16}$$

where $P_{radius,M}(r) = P_{radius}(r)$ (*Figure 2—figure supplement 1F*) and $P_{radius,D}(r)$ is the re-normalized distribution after setting the probability to zero for r bins < 25 nm (*Figure 2—figure supplement 1G*). Note, the total number of liposomes does not change in the dimer simulation, but the small r matrices remain empty, contributing to the fraction of unoccupied vesicles, $F_0$.

$N_{protein,M}(r)$ and $N_{protein,D}(r)$ – the number of monomers or dimers that are inserted into $N_{liposomes,M}(r)$ and $N_{liposomes,D}(r)$.

For a liposome population with radius r, the number of protein oligomers that will be inserted into those liposomes depends on the fraction of the total membrane surface area in that liposome population. Note that different accessible liposome distributions must be used for the monomer and dimer to account for the observed experimental $F_0$.

$$N_{protein,M}(r) = \left( \frac{r^2 P_{radius,M}(r)}{\sum_r r^2 P_{radius,M}(r)} \right) \cdot N_{protein,M} \qquad (17)$$

$$N_{protein,D}(r) = \left( \frac{r^2 P_{radius,D}(r)}{\sum_r r^2 P_{radius,D}(r)} \right) \cdot N_{protein,D} \qquad (18)$$

The simulation follows by inserting the appropriate number of the various monomer fluorescent species, $N_{protein,M}(r) * P_{label,M}(N_{Cy5})$, into $N_{liposomes,M}(r)$. Similarily, $N_{protein,D}(r) * P_{label,D}(N_{Cy5})$ is used to calculate the number of the different dimer fluorescent species to be inserted into $N_{liposomes,D}(r)$. After the simulation, the total number of Cy5 molecules is counted in each liposome, and a histogram is calculated over the total liposome population. The histogram is normalized by $N_{liposomes}$ to yield the fluorophore occupancy probability distribution $P*$. Therefore, the fraction of unoccupied liposomes is:

$$F_0 = P^*(0) \qquad (19)$$

The probability of Cy5 occupied liposomes that have one or two Cy5 fluorophores is calculated as:

$$P_1 = \frac{P^*(1)}{1 - P^*(0)} \qquad (20)$$

$$P_2 = \frac{P^*(2)}{1 - P^*(0)} \qquad (21)$$

$$P_{3+} = \frac{1 - P^*(0) - P^*(1) - P^*(2)}{1 - P^*(0)} \qquad (22)$$

which corresponds to the experimental photobleaching probabilities described in equations 1–3. Note that the MATLAB script is available for download in the Supplemental section entitled - Source Code.

## Determining the dimerization equilibrium constant and standard state free energy from the experimental photobleaching data.

A brief description of the derivation of the dimerization equilibrium isotherm is provided here (*Wyman and Gill, 1990*). The equilibrium dimerization reaction scheme can be written as follows for monomers, M, and dimers, D:

$$M + M \rightleftharpoons D \qquad (23)$$

Membrane proteins react in a two-dimensional lipid bilayer and so the scale of the reaction coordinate must be appropriately selected. As discussed in the literature (*White and Wimley, 1994*; *1999*; *Fleming, 2002*; *Zhang and Lazaridis, 2006*), the subunit/lipid mole fraction scale, $\chi$, is a conventional choice for studying membrane protein association, as it can apply to reactions in both detergent micelles and lipids. For reactions in lipid bilayers, the area density scale, $\rho_{Area}$ (subunit/nm$^2$) is intuitive and often used (*Hong et al., 2010*). In the end, both scales are imperfect and require further corrections to account for the differences in the sizes of subunit and lipid solvent molecules (*White and Wimley, 1994*; *Fleming, 2002*). For consistency with the current literature, we report the equilibrium constant $K_\chi$, dissociation constant $Kd$ and standard state free energy $\Delta G°$ on the subunit/lipid mole fraction scale (*Table 2*), but also include the subunit/nm$^2$ the area density

**Table 2.** Summary of dissociation constants, equilibrium association constants and standard state free energy based on the best-fit parameters of $F_{Dimer}$ vs. the reactive mole fraction, $\chi^*$ or area density, $\rho_{Area}^*$.

| ClC-ec1 construct | | Mole fraction scale ($\chi^*$) standard state = 1 subunit/lipid | | | Area density scale ($\rho_{Area}^*$) standard state = 1 subunit/nm$^2$ | | | |
|---|---|---|---|---|---|---|---|---|
| | | $K_d$ (subunits/ lipid) | $K_\chi$ (lipids/ subunit) | $\Delta G°\chi$ (kcal/mole) | $K_d$ (subunits/ nm$^2$) | $K_\rho$ (nm$^2$/ subunit) | Eq. Box (nm × nm) | $\Delta G°\rho$ (kcal/mole) |
| WT | mean ± SE | $4.7 \pm 1.1 \times 10^{-9}$ | $2.1 \pm 0.5 \times 10^{8}$ | $-11.4 \pm 0.1$ | $1.6 \pm 0.4 \times 10^{-8}$ | $6.4 \pm 1.4 \times 10^{7}$ | $8002 \pm 3794$ | $-10.7 \pm 0.1$ |
| $Y_0$ const. = 0.07 | 95% CI | $2.6$ to $6.8 \times 10^{-9}$ | $1.2$ to $3.1 \times 10^{8}$ | $-11.6$ to $-11.1$ | $8.5 \times 10^{-9}$ to $2.3 \times 10^{-8}$ | $3.5$ to $9.3 \times 10^{7}$ | $5919$ to $9644$ | $-10.9$ to $-10.4$ |
| W | mean ± SE | $2.7 \pm 1.1 \times 10^{-7}$ | $3.7 \pm 1.6 \times 10^{6}$ | $-9.0 \pm 0.3$ | $8.9 \pm 3.8 \times 10^{-7}$ | $1.1 \pm 0.5 \times 10^{6}$ | $1049 \pm 707$ | $-8.3 \pm 0.3$ |
| $Y_0 = 0.07 \pm 0.06$ | 95% CI | $3.5 \times 10^{-8}$ to $5.0 \times 10^{-7}$ | $2.3$ to $5.1 \times 10^{6}$ | $-9.5$ to $-8.5$ | $1.2 \times 10^{-7}$ to $1.7 \times 10^{-6}$ | $1.5 \times 10^{5}$ to $2.1 \times 10^{6}$ | $387$ to $1449$ | $-8.8$ to $-7.7$ |
| WW | mean ± SE | $4.7 \pm 1.8 \times 10^{-6}$ | $2.1 \pm 0.8 \times 10^{5}$ | $-7.3 \pm 0.2$ | $1.6 \pm 0.6 \times 10^{-5}$ | $6.3 \pm 2.4 \times 10^{4}$ | $251 \pm 155$ | $-6.6 \pm 0.2$ |
| $Y_0 = 0.06 \pm 0.02$ | 95% CI | $1.2$ to $8.3 \times 10^{-6}$ | $5.0 \times 10^{4}$ to $3.7 \times 10^{5}$ | $-7.7$ to $-6.8$ | $3.9 \times 10^{-6}$ to $2.8 \times 10^{-5}$ | $1.5 \times 10^{4}$ to $1.1 \times 10^{5}$ | $122$ to $332$ | $-7.0$ to $-6.1$ |

Best-fit parameters are reported as mean ± standard error (SE) and 95% confidence intervals (CI). The area density scale is calculated by converting the mole fraction scale using $SA_{lipid}$ = 0.6 nm$^2$ per lipid in a single leaflet and is not corrected for differences in the subunit vs. lipid volume. Eq. Box denotes the box size defined by the equilibrium constant $K_{eq}$. $Y_0$ indicates the baseline offset parameter that is either fitted or constrained.

scale for reference. The area density scale is calculated from the mole fraction scale using the following equation:

$$\rho_{Area} = \frac{2\chi}{SA_{lipid}} \tag{24}$$

where $SA_{lipid}$ is the surface area per lipid in nm$^2$. Assuming ideal dilute conditions, the mole fraction equilibrium constant for dimerization is:

$$K_{\chi^*} = \frac{\chi_D^*}{(\chi_M^*)^2} \tag{25}$$

where $\chi^*$ is the total reactive mole fraction, equivalent to $\chi/2$, that subunits are randomly incorporated into the membrane (*Matulef and Maduke, 2005*) and that the reaction only occurs between oriented subunits. $\chi_D^*$ is the dimer/lipid mole fraction, $\chi_M^*$ is the monomer/lipid mole fraction, and $K_{\chi}^*$ is the dimerization equilibrium constant in inverse mole fraction units (i.e. lipid/subunit) (*Fleming, 2002*). The total mole fraction of subunits in the membrane is:

$$\chi^* = \chi_M^* + 2\chi_D^* \tag{26}$$

The fraction of protein in the dimer state ($F_{Dimer}$) is derived by substituting (26) into (25) to determine $\chi_D$:

$$F_{Dimer} = \frac{2\chi_D^*}{\chi^*} = \frac{1 + 4\chi^* K_{\chi^*} - \sqrt{1 + 8\chi^* K_{\chi^*}}}{4\chi^* K_{\chi^*}} \tag{27}$$

To determine $F_{Dimer}$ from the photobleaching analysis, we used the theoretical calculations of the ideal monomer probabilities, $P^D_{n=1-5+}$, and ideal dimer probabilities, $P^D_{n=1-5+}$ as the respective all-monomer and all-dimer signals. Least-squares analysis was carried out on the sum of squared residuals ($R^2$) between the experimental data, $P^{expt}_n$, and a linear combination of $P^M_{n=1-5+}$, and, $P^D_{n=1-5+}$, weighted by $F_{Dimer}$

$$R^2 = \sum_{n=1-5+} \left(P^{expt}_n - \left((1 - F_{Dimer}) \cdot P^M_n + F_{Dimer} \cdot P^D_n\right)\right)^2 \tag{28}$$

The minimum $R^2$ value corresponds to the predicted $F_{Dimer}$ for a given photobleaching distribution (*Figure 5—figure supplement 1*). $F_{Dimer}$ vs. $\chi^*$ was fit to the equilibrium dimerization isotherm above with the addition of a baseline offset $Y_0$:

$$F_{Dimer} = (1 - Y_0) * \left( \frac{1 + 4\chi^* K_{\chi^*} - \sqrt{1 + 8\chi^* K_{\chi^*}}}{4\chi^* K_{\chi^*}} \right) + Y_0 \qquad (29)$$

Analysis of the $R^2$ as a function of mole fraction density shows that the quality of the fits of the experimental distributions to the theoretical distributions deviate for $\chi^* > 1.9 \times 10^{-6}$ subunits/lipid (*Figure 5—figure supplement 2*), which could arise from inaccuracies in the liposome size distribution by exclusion of larger liposomes or multilamellar liposomes, or a small proportion of non-specific oligomerization. To account for this uncertainty in $F_{Dimer}$ estimation at high densities, we weighted the fits by $1/R^2$, essentially limiting the data to a dynamic range of $\chi^* = 3.8 \times 10^{-6}$ to $1.9 \times 10^{-6}$. The weighted non-linear fits were carried out using the non-linear regression function in MATLAB.

The mole fraction standard state free energy is calculated as:

$$\Delta G^\circ = -RT \, ln\left( \chi^\circ K_{\chi^*} \right) \qquad (30)$$

where $R$ is the gas constant (1.987 2036 cal/mole K), T is the temperature (~298 K) and $\chi^\circ$ is the standard state mole fraction density 1 subunit/lipid. Note that this standard state is not physically relevant, but provides a normalization point for other measurements to be compared. On the area density scale, the standard state density $\rho^\circ = 1$ subunit/nm$^2$ is used. For comparison, the typical standard state of 1 M on the molar scale is equivalent to 1 subunit/1.6 nm$^3$ or 1 subunit/54 water molecules assuming a molecular volume of water of 30 Å$^3$ (*White and Wimley, 1999*)

## Acknowledgements

We are extremely grateful to Christopher Miller for constructive discussions throughout the project, as well as Nicholas Last, and the Miller and Gelles laboratories for feedback early on. We would also like to thank Laura Hughes, Bob Rawle and Steven Boxer for advice on fluorophores and introduction to the 'splat' approach, and to Benoît Roux for useful feedback about the manuscript. The research was supported by NIH/NIGMS grant R00GM101016 and a Roy J. Carver Trust Foundation Early Investigator Award.

## Additional information

### Funding

| Funder | Grant reference number | Author |
|---|---|---|
| National Institutes of Health | R00GM101016 | Venkatramanan Krishnamani<br>Kacey Mersch<br>Janice L Robertson |
| Roy J. Carver Charitable Trust | Early Investigator Award | Rahul Chadda<br>Marley Brimberry<br>Ankita Chadda<br>Janice L Robertson |

The funders had no role in study design, data collection and interpretation, or the decision to submit the work for publication.

### Author contributions

RC, VK, KM, JW, MB, AC, LK-P, Acquisition of data, Analysis and interpretation of data, Drafting or revising the article; LJF, JLR, Conception and design, Acquisition of data, Analysis and interpretation of data, Drafting or revising the article; JG, Conception and design, Analysis and interpretation of data, Drafting or revising the article

**Author ORCIDs**

Larry J Friedman, http://orcid.org/0000-0003-4946-8731

Jeff Gelles, http://orcid.org/0000-0001-7910-3421

Janice L Robertson, http://orcid.org/0000-0002-5499-9943

## Additional files

**Supplementary files**

• Source code 1. DimerizationRobertson2016v1.m – MATLAB script that calculates the theoretical monomer and dimer photobleaching distribution $P_{1-5+}$.

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
