## [Decision Letter]

Thank you for submitting your article "The dimerization equilibrium of a ClC Cl^-^/H^+^ antiporter in lipid bilayers" for consideration by *eLife*. Your article has been favorably evaluated by Richard Aldrich as the Senior editor and three reviewers, including Karen Fleming (Reviewer #3) and Olga Boudker (Reviewer #1), who is a member of our Board of Reviewing Editors.

The reviewers have discussed the reviews with one another and the Reviewing Editor has drafted this decision to help you prepare a revised submission. We hope you will be able to submit the revised version within two months, so please let us know if you have any questions first.

Summary:

In the manuscript, Robertson et al. describe a method designed to measure association energy of ClC dimer. The method is based on exceptionally high dilution of the dimers in model membranes, partitioning of membranes into vesicles and protein counting in the vesicles using single molecule TIRF microscopy. Coupled with careful statistical considerations, the method allows for the determination of self-association isotherms and thus, free energies. This method can now be applied to better understand thermodynamics of protein:protein interactions in the bilayer.

Essential revisions:

1) The protein-to-lipid molar ratio (χ) or protein density appears to represent their initial mixing ratio before reconstitution (dialysis). However, the multiple steps towards final extruded samples (i.e., dialysis, gel filtration, and freeze/thaw cycles) may lead to a significant deviation of the actual χ values from the initial numbers due to a loss of lipids and protein aggregation/denaturation in aqueous/lipid phases. The valid χ values for obtaining DG should be those after freeze/thaw cycles and before extrusion (i.e., the equilibration step). Therefore, the reliability of the χ values needs to be validated.

2) In the subsection “ClC-ec1 follows a monomer to dimer reaction as a function of protein density in the membrane” and Figure 3: To verify the equilibration of the samples, the authors used W-Cy5, the result of which was very convincing. However, it is also possible that WT-Cy5 dimers may have a high kinetic barrier to dissociation (related reference: Jefferson, Blois and Bowie, J. Am. Chem. Soc., 2013, 135, 15183-15190). It is desirable that the equilibration is confirmed for WT-Cy5 as well.

3) In the last paragraph of the Results section and Figure 5: The isotherm data in Figure 5 look very good. Why did the authors not estimate the errors in ΔG? The free energy changes should be presented and discussed as a number with error bars. Stating that the free energy of dimerization is ~-11 kcal/mole diminishes the significance of the work and makes it impossible to know if the W mutants that are 2 kcal/mole destabilized is a significant difference or not. Even if the error bar is derived from the error of the fit (because each point has a counting error), this should still be information on the precision of the measurement. Because the investigators have a method of directly knowing the stoichiometry, they know the curve shape, and we think it is fine to only access only a part of the isotherm (as in the WW data in Figure 5) and still derive some kind of uncertainty on the measurement. One type of analysis that could be implemented is a bootstrap with replacement method to clarify the error space?

4) One aspect that seems worth additional discussion is conversion of equilibrium constants expressed in terms of subunit/lipid density χ into standard Gibbs free energy of interaction. It would be helpful if the authors provided a brief discussion on the subject and included relevant references.

5) Figure 1—figure supplement 2: To verify the functional integrity of labeled and mutated samples, the functional assay result for WT-Cy5 should be presented.

6) One aspect that is not clear as written is how the "actual state of the protein in the final liposome is irrelevant to this analysis". Is this not the state where monomers and dimers are imaged and counted? This needs to be better explained. Also, if all ClC proteins will be dimers in *E. coli* as stated in the Discussion, it is an apparent contradiction to find monomers in vesicles with such favorable dimerization energy. An alternative explanation is that the kinetics of monomer/dimer rearrangement in the small vesicles is so slow as to be frozen. And if that is the case, why is that and will that be true for other lipids?

---

## [Author Response]

*Essential revisions:*

*1) The protein-to-lipid molar ratio (χ) or protein density appears to represent their initial mixing ratio before reconstitution (dialysis). However, the multiple steps towards final extruded samples (i.e., dialysis, gel filtration, and freeze/thaw cycles) may lead to a significant deviation of the actual χ values from the initial numbers due to a loss of lipids and protein aggregation/denaturation in aqueous/lipid phases. The valid χ values for obtaining DG should be those after freeze/thaw cycles and before extrusion (i.e., the equilibration step). Therefore, the reliability of the χ values needs to be validated.*

Thank you for raising this important point. We measured the subunit/lipid mole fraction for samples (old and newly prepared) after freeze/thaw and extrusion and have included these results in Figure 1—figure supplement 3. We measured the mole fraction as 0.50 ± 0.02 of the mole fraction set during reconstitution, independent of protein density or construct. We have adjusted all of the results in relation to the observed mole fraction measured by this analysis. A section has been added to the Methods (” Protein/lipid quantification”) describing our method of protein quantification (comparing absolute fluorescence of denatured protein to a standard curve), and lipid quantification by measurement of phosphate.

*2) In the subsection “ClC-ec1 follows a monomer to dimer reaction as a function of protein density in the membrane” and Figure 3: To verify the equilibration of the samples, the authors used W-Cy5, the result of which was very convincing. However, it is also possible that WT-Cy5 dimers may have a high kinetic barrier to dissociation (related reference: Jefferson, Blois and Bowie, J. Am. Chem. Soc., 2013, 135, 15183-15190). It is desirable that the equilibration is confirmed for WT-Cy5 as well.*

The data for the WT-Cy5 dilution experiment is now presented in Figure 3. As was observed for W-Cy5, 10X and 100X dilutions of WT shows a complete shift to their reconstituted distribution 1 day after freeze/thaw. We also carried out a 1000X dilution that is expected to significantly shift the WT population towards the monomeric state. While the diluted distribution showed a shift towards monomer, it was significantly different from the dialysis samples reconstituted at the same mole fraction density. We measured the time course of the 1000X WT dilution over the course of 25 days showing that this sample converges to the probability distribution of samples reconstituted at χ = 3.8 x 10^-9^ subunit/lipid, with a mid point of 13 days. This indicates that the reconstituted WT samples do reflect the protein population at equilibrium in the membrane, and that WT has high kinetic stability compared to W, as the reviewers indicated.

*3) In the last paragraph of the Results section and Figure 5: The isotherm data in Figure 5 look very good. Why did the authors not estimate the errors in DG? The free energy changes should be presented and discussed as a number with error bars. Stating that the free energy of dimerization is ~-11 kcal/mole diminishes the significance of the work and makes it impossible to know if the W mutants that are 2 kcal/mole destabilized is a significant difference or not. Even if the error bar is derived from the error of the fit (because each point has a counting error), this should still be information on the precision of the measurement. Because the investigators have a method of directly knowing the stoichiometry, they know the curve shape, and we think it is fine to only access only a part of the isotherm (as in the WW data in Figure 5) and still derive some kind of uncertainty on the measurement. One type of analysis that could be implemented is a bootstrap with replacement method to clarify the error space?*

We have added Table 1 reporting the standard state free energy, equilibrium and dissociation constants and error values. These represent best-fit values ± standard error of the fits from direct fitting of each parameter, as well as the 95% confidence intervals, obtained from non-linear weighted regression implemented in MATLAB. We have also added a column where the mole fraction scale has been converted to an area density scale, for easy comparison with literature. We have also elaborated on these values in the last paragraph in the Results section.

*4) One aspect that seems worth additional discussion is conversion of equilibrium constants expressed in terms of subunit/lipid density χ into standard Gibbs free energy of interaction. It would be helpful if the authors provided a brief discussion on the subject and included relevant references.*

A section has been added to the methods titled “Determining the dimerization equilibrium constant and standard state free energy from the experimental photobleaching data”. In this section, we describe the describe the derivation of the dimerization isotherm for determination of the equilibrium constant, choice of mole fraction vs. area density reaction scales and calculation of the standard state free energy with the conventional standard states.

*5) Figure 1—figure supplement 2: To verify the functional integrity of labeled and mutated samples, the functional assay result for WT-Cy5 should be presented.*

This data is now reported in Figure 1—figure supplement 2. WT-Cy5 is functional, and there is no significant difference in the fraction of occupied liposome volume *F_Cl,0_*.

*6) One aspect that is not clear as written is how the "actual state of the protein in the final liposome is irrelevant to this analysis". Is this not the state where monomers and dimers are imaged and counted? This needs to be better explained.*

Thank you for pointing this out. This is a key point of the paper and so we would like this to be clear to the readers. We have modified the text throughout the paper to clarify this, including the following paragraph at the beginning of the Results section:

“With single-molecule resolution, we can count the number of subunits captured into each liposome (Figure 1) and determine the photobleaching probability distribution of fluorescent protein occupancy in liposomes (Figure 1). […] Therefore, the liposome extrusion step captures the monomer-dimer equilibrium in the prior MLV membrane state, and ignores any changes in protein density or lipid composition that might arise during the extrusion process.”

*Also, if all ClC proteins will be dimers in E. coli as stated in the Discussion, it is an apparent contradiction to find monomers in vesicles with such favorable dimerization energy.*

The following paragraph in the Discussion has been modified for clarification:

“Our lowest experimental density corresponds to 1 subunit per 50 *E. coli* inner membranes, assuming a 4 µm^2^ surface area consisting of ~10^7^ lipids (Prats & de Pedro 1989). […] This means that any reasonable level of expression (10-100 copies per cell) will drive the reaction into a range where the protein will exist mainly as dimers, with negligible probability of observing the dissociated monomeric state.”

*An alternative explanation is that the kinetics of monomer/dimer rearrangement in the small vesicles is so slow as to be frozen. And if that is the case, why is that and will that be true for other lipids?*

Since our method does not report on the actual state of the protein in the small, extruded vesicles – only the number of fluorescently labeled subunits captured into these vesicles – we cannot comment on the kinetics of subunits in the liposomes. This would be something that would require a different approach, such as dynamic FRET, and does not pertain to our measurements of the protein in a large bilayer at equilibrium. In addition, we will not comment on the impact of other lipid compositions as this is something that must be measured directly and is outside the scope of the current study.